# TRPC3 shapes the ER-mitochondria Ca$^{2+}$ transfer characterizing tumour-promoting senescence

Valerio Farfariello [1,2,11,12✉], Dmitri V. Gordienko[1,2,11], Lina Mesilmany [1,2,3,11], Yasmine Touil [4], Emmanuelle Germain[1,2], Ingrid Fliniaux[1,2], Emilie Desruelles[1,2], Dimitra Gkika[1,2,4], Morad Roudbaraki[1,2], George Shapovalov[1,2], Lucile Noyer [5], Mathilde Lebas[1,2], Laurent Allart[1,2], Nathalie Zienthal-Gelus[1,2], Oksana Iamshanova[6], Franck Bonardi[7], Martin Figeac[7], William Laine[4], Jerome Kluza[4], Philippe Marchetti[4], Bruno Quesnel [4], Daniel Metzger [8], David Bernard [9], Jan B. Parys [10], Loïc Lemonnier [1,2,12] & Natalia Prevarskaya [1,2,12✉]

Cellular senescence is implicated in a great number of diseases including cancer. Although alterations in mitochondrial metabolism were reported as senescence drivers, the underlying mechanisms remain elusive. We report the mechanism altering mitochondrial function and OXPHOS in stress-induced senescent fibroblasts. We demonstrate that TRPC3 protein, acting as a controller of mitochondrial Ca$^{2+}$ load via negative regulation of IP$_3$ receptor-mediated Ca$^{2+}$ release, is down regulated in senescence regardless of the type of senescence inducer. This remodelling promotes cytosolic/mitochondrial Ca$^{2+}$ oscillations and elevates mitochondrial Ca$^{2+}$ load, mitochondrial oxygen consumption rate and oxidative phosphorylation. Re-expression of TRPC3 in senescent cells diminishes mitochondrial Ca$^{2+}$ load and promotes escape from OIS-induced senescence. Cellular senescence evoked by TRPC3 downregulation in stromal cells displays a proinflammatory and tumour-promoting secretome that encourages cancer epithelial cell proliferation and tumour growth in vivo. Altogether, our results unravel the mechanism contributing to pro-tumour behaviour of senescent cells.

[1] Université de Lille, Inserm, U1003 - PHYCEL - Physiologie Cellulaire, F-59000 Lille, France. [2] Laboratory of Excellence, Ion Channels Science and Therapeutics, Villeneuve d'Ascq, France. [3] Laboratory of Cancer Biology and Molecular Immunology, Faculty of Sciences, Lebanese University, Hadat, Beirut, Lebanon. [4] Université de Lille, CNRS, Inserm, CHU Lille, UMR9020-U1277 - CANTHER - Cancer Heterogeneity Plasticity and Resistance to Therapies, F-59000 Lille, France. [5] Department of Pathology, NYU Langone Medical Center, New York City, USA. [6] Institute of Biochemistry and Molecular Medicine, University of Bern, Bern, Switzerland. [7] Université de Lille, CHU Lille, Plate-forme de génomique fonctionnelle, Centre de Biologie et Pathologie, F-59000 Lille, France. [8] Department of Functional Genomics and Cancer, Institut de Génétique et de Biologie Moléculaire et Cellulaire (IGBMC), Centre National de la Recherche Scientifique UMR7104/Institut National de la Santé et de la Recherche Médicale U1258, Université de Strasbourg, Strasbourg, Illkirch Cedex, France. [9] Inserm U1052, Centre de Recherche en Cancérologie de Lyon, CNRS 5286, Centre Léon Bérard, Université de Lyon, Lyon, France. [10] KU Leuven, Laboratory of Molecular and Cellular Signaling, Department of Cellular and Molecular Medicine & Leuven Kanker Instituut, Leuven, Belgium. [11] These authors contributed equally: Valerio Farfariello, Dmitri V. Gordienko, Lina Mesilmany. [12] These authors jointly supervised: Valerio Farfariello, Loïc Lemonnier, Natalia Prevarskaya. ✉email: valerio.farfariello@inserm.fr; natacha.prevarskaya@univ-lille.fr

It is now well established that various cell-stressing factors, such as DNA-damaging radiations, oxidative stress or onco-gene activation, are able to initiate an alternative cellular response to DNA damage that engages the p53 and pRB tumour suppressor pathways culminating in the induction of premature senescence[1–3]. Senescent cells are depicted by specific character-istics such as morphological modifications, cell cycle arrest, apoptosis resistance and altered gene expression. Despite the irreversible loss of their proliferative ability, senescent cells remain metabolically active and secrete several soluble factors,

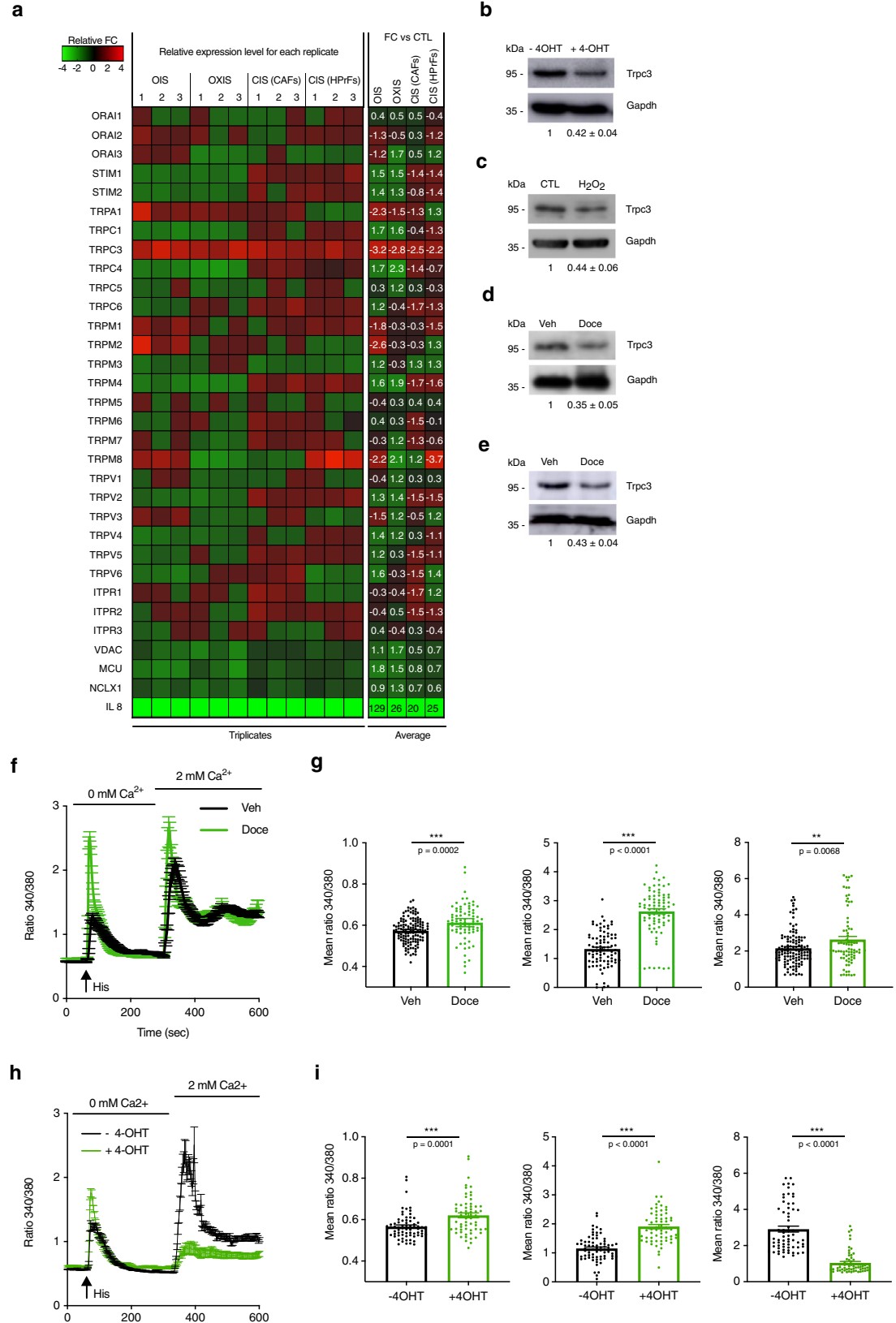

**Fig. 1 Senescent prostate stromal cells have an altered calcium homoeostasis. a** Heatmap of mRNA levels for the indicated ion channels and accessory proteins after chemotherapy-induced senescence in HPrFs or CAFs, oxidative stress-induced senescence in HPrFs and oncogene-induced senescence in MRC5 fibroblasts. Mean values of the fold changes relative to controls ($n = 3$) are presented in red-green colour scale (see scalebar). IL-8 mRNA expression was used as a positive control for senescence induction. **b–e** TRPC3 protein levels are shown in the four models of senescence. Data are expressed as mean ± SEM ($n = 3$). Western blot images are representative of three independent experiments. **f–i** Cytosolic Ca$^{2+}$ imaging (Fura-2) was performed on HPrFs (CIS, Docetaxel, 5 nM) or MRC5 (OIS, 4-OHT, 250 nM). **f** and **h** Averaged time courses (mean ± SEM) of fluorescence ratio with excitation wavelengths 340/380 nm. **g** and **i** Means values of (1) basal cytosolic Ca$^{2+}$ levels, (2) peak amplitudes of the Histamine (100 μM)-induced Ca$^{2+}$ release from the ER and (3) peak amplitudes of SOCE, respectively ($n_{Veh} = 212$, $n_{Doce} = 234$; $n_{-4OHT} = 66$, $n_{+4OHT} = 61$. Data are presented as mean ± SEM ($n = 3$). ***$P < 0.001$ (Student's $t$ test, two-sided). See also Supplementary Fig. 1.

referred to as senescence-associated secretory phenotype (SASP)[4–6], which includes secreted proteins like metalloproteases, inflammatory cytokines and growth factors.

Chemotherapeutic agents have also been shown to induce so-called chemotherapy-induced senescence (CIS)[1–3]. The induction of CIS in the tumour microenvironment alters the cross-talk between tumour and stromal cells via development of the SASP, thus supporting tumour relapse and the resistance to further antineoplastic therapies. Indeed, stromal cell CIS is able to boost neoplastic epithelial cell proliferation by paracrine mechanisms. This involves molecules such as FGF, HGF, amphiregulin and WNT16B recently identified as components of the SASP in the tumour microenvironment and capable of promoting tumour cell survival/disease progression and attenuating the effects of cytotoxic chemotherapy[7–10]. It is therefore vital to identify molecular actors regulating senescence in stromal cells and the underlying molecular mechanisms.

In recent years, emerging evidence indicate that the mitochondrial activity is coupled to cellular senescence. Mitochondrial dysfunction has been described to induce a specific subset of senescence called "mitochondrial dysfunction-associated senescence (MiDAS[11]) and more recently, NAD$^+$ metabolism has been related to the acquisition of a proinflammatory senescence-associated secretome[12]. In addition, mitochondrial reactive oxygen species (ROS) production is known to induce/stabilise the DNA damage response (DDR), leading to a permanent growth arrest[13]. It is now clear that mitochondrial metabolism is tightly coupled to cellular Ca$^{2+}$ homoeostasis. Indeed, mitochondria both encode and decode Ca$^{2+}$ signals which in turn have a direct impact on oxidative phosphorylation (OXPHOS) and ATP production. Indeed, mitochondrial Ca$^{2+}$ transients have been reported to increase mitochondrial dehydrogenase (DHase) activity that in turn enhances ATP synthesis. Taking into account that three key DHases of the tricarboxylic acid (TCA) cycle are Ca$^{2+}$ dependent, it was suggested that mitochondrial Ca$^{2+}$ regulates the NADH/NAD+ ratio and, hence, controls ATP synthesis[14–16].

Nevertheless, the mechanisms underlying mitochondrial Ca$^{2+}$ homoeostasis alteration in senescent cells are not yet well understood. Normal mitochondrial Ca$^{2+}$ homoeostasis relies on functional endoplasmic reticulum (ER)–mitochondrial Ca$^{2+}$ transfer due to constitutive activity of inositol 1,4,5-trisphosphate receptors (IP$_3$R)[17]. We and others have recently shown that IP$_3$R-mediated Ca$^{2+}$ release from the ER and subsequent Ca$^{2+}$ accumulation in mitochondria lead to cellular senescence in normal human cells[18–21]. It is important to note that a great number of partner proteins not only regulate (either directly or via multiple signalling pathways) the IP$_3$R activity, but also determine the molecular targets of IP$_3$R-mediated Ca$^{2+}$ release[22]. Among these, a number of oncogenes and tumour suppressors proteins have been identified[23–25] shedding light on the Ca$^{2+}$ dependence of the cancer hallmarks. Whether alteration of the IP$_3$R activity in senescence is mediated by partner protein(s), however, remains unknown.

Here we show how cytosolic and mitochondrial Ca$^{2+}$ signatures are altered in senescent cells and whether these alterations affect the SASP composition and tumour progression. Our results provide original insights into the determinants of Ca$^{2+}$-dependence in pro-tumour senescence.

## Results

**Senescent cells display TRPC3 downregulation and altered calcium homoeostasis.** One of the most widespread families of Ca$^{2+}$-permeable channels known to affect IP$_3$R activity and to be recruited during capacitative Ca$^{2+}$ entry are transient receptor potential (TRP) channels[22,26].

In order to assess possible changes in the expression of Ca$^{2+}$-permeable channels in senescence, we used several experimental models: (1) human prostate fibroblasts (HPrFs) and prostate cancer-associated fibroblasts (CAFs) treated with docetaxel (Doce)—a microtubule poison commonly used in prostate cancer therapy[27,28] and known to induce senescence in several cell types[3,9]; (2) HPrFs treated with hydrogen peroxide, H$_2$O$_2$[29,30] and (3) MRC-5 fibroblasts in which senescence was induced by oncogenic RAS (hRAS$^{G12V}$ [31]). The induction of senescence was confirmed by SA-β-galactosidase staining (Supplementary Fig. 1a–d)[32].

Knowing that, on the one hand, Ca$^{2+}$-permeable channels are involved in the establishment of senescence[18] and, on the other hand, a number of tumours are considered as oncochannelopathies[33], we therefore performed a real-time qPCR screening for the members of TRP, IP$_3$Rs, ORAI channel families and the accessory molecules STIM and some of the main Ca$^{2+}$-permeable mitochondrial channels. As shown by the heat map in Fig. 1a, the senescent phenotype was associated with alterations in the expression of several genes, among which *Trpc3* was the only gene found to be consistently and significantly downregulated in all models of senescence (Supplementary Table 2). This was also confirmed at the protein level (Fig. 1B–E). Moreover, the decline of *Trpc3* expression was found to occur at an early stage of the senescent phenotype establishment (Supplementary Fig. 1e–h). Noteworthily, the expression of the genes encoding diacylglycerol (DAG)-sensitive TRPC3 homologues, *Trpc6* and *Trpc7* (the latter was not expressed) was not affected.

We analysed the expression of TRPC3 in human prostate cancer tissues using published microarray datasets obtained from chemo/radiotherapy-treated patients. We found that TRPC3, reported to be predominantly expressed in the prostate stroma[34], was downregulated in tumours from patients undergoing anticancer treatments (Supplementary Fig. 1j–k) (ONCO-MINE: Datasets "Grasso Prostate" and "Tamura Prostate"). This was also true for other malignancies, such as breast cancer (Supplementary Fig. 1l). TRPC3 downregulation in senescent cells was also confirmed by the analysis of public datasets reporting gene expression changes caused by treatments with pro-senescence compounds or RAS expression, as depicted in Supplementary Fig. 1l.

Moreover, in the tumours expressing low levels of TRPC3, the gene set enrichment analysis (GSEA) conducted on the data from the gene expression omnibus (GEO) dataset GSE35988 from prostate cancer tissues showed a strong enrichment of the genes upregulated in senescence (Human Cellular Senescence Gene Database, HCSGD[35], Supplementary Fig. 1m).

To decipher the $Ca^{2+}$ signalling events characterising senescence, basal cytosolic $Ca^{2+}$ levels, the magnitudes of $Ca^{2+}$ release and $Ca^{2+}$ entry were measured in two senescence models, CIS and oncogene-

induced senescence (OIS), utilising ratiometric fluorescence wide-field $Ca^{2+}$ imaging with Fura-2/AM (Fig. 1f–i). Our results show a slight but significant increase in basal cytosolic $Ca^{2+}$ concentration ($[Ca^{2+}]_c$) in senescent cells. Application of histamine (an endogenous activator of the G-protein–phospholipase C–$IP_3$–$IP_3R$-mediated $Ca^{2+}$ release signalling pathway[36]) to cells bathed in a $Ca^{2+}$-free solution revealed that $Ca^{2+}$ release from the ER is dramatically increased in senescent cells. This was associated with a significant increase in capacitative $Ca^{2+}$ entry (SOCE) following $Ca^{2+}$

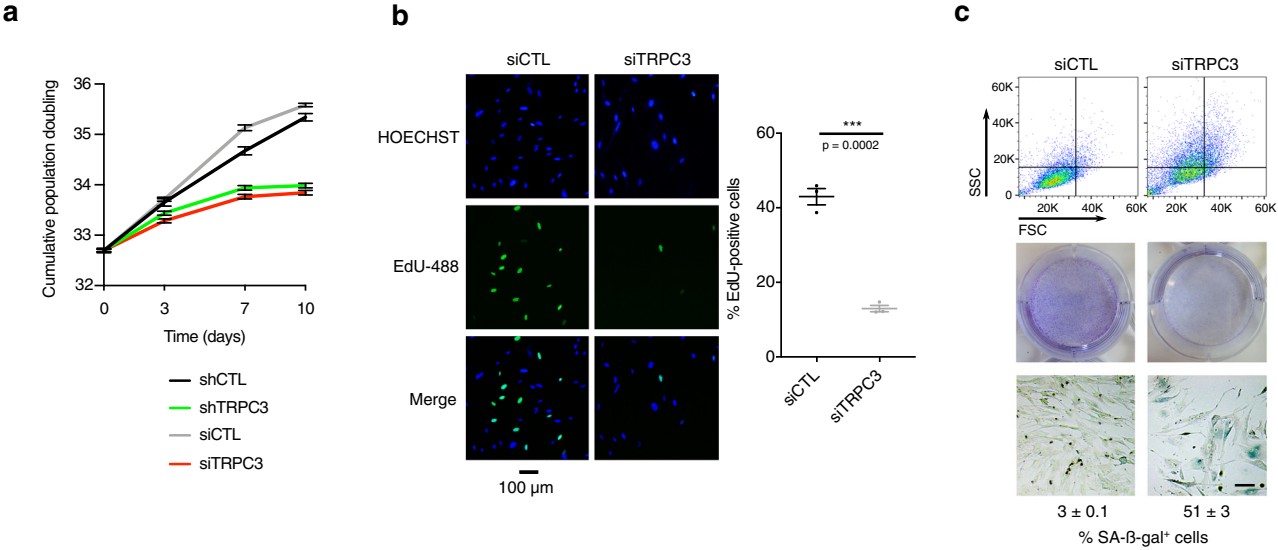

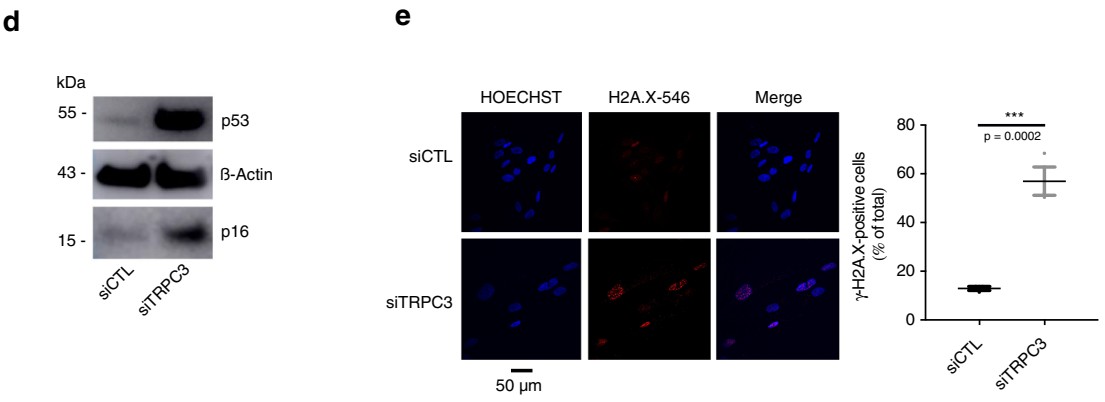

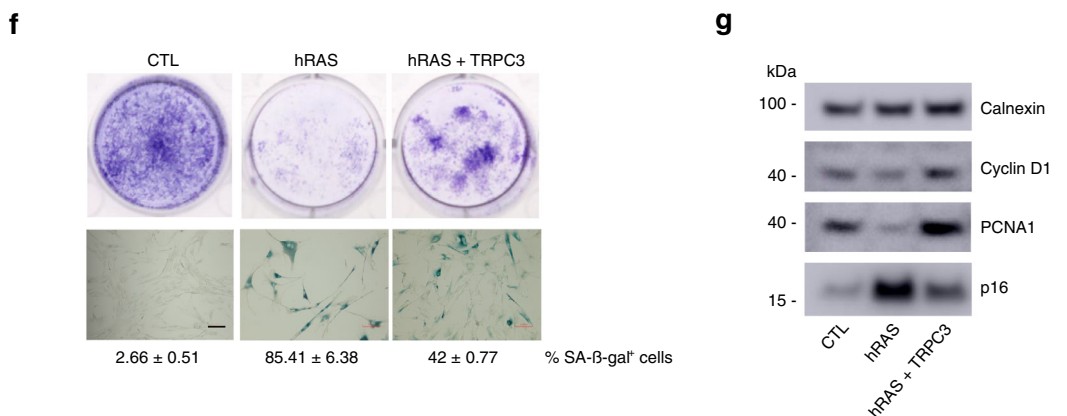

**Fig. 2 TRPC3 controls senescence. a** Cumulative population doublings (PD) of HPrFs either stably expressing doxycycline-inducible shRNA against TRPC3 or treated with siTRPC3 and their siControls (siCTL, scrambled sequences). Results are presented as mean ± SEM; n = 3). **b** EdU staining on MRC5 cells treated 10 days with siCTL or siTRPC3. % of EdU positive cells is shown (mean ± SEM; n = 3). Scale bar = 150 μm. ***P < 0.001 (Student's t test, two-sided). **c** Effect of siRNA-mediated TRPC3 knockdown in HPrFs on: (Top) the cell structural parameters revealed with flow cytometry analysis—cell size (forward scatter; FSC) and internal complexity (side scattered, SSC), (Middle) cell growth reported by crystal violet staining and (Bottom) β-gal activity revealed with SA-β-gal staining (% of ß-galactosidase positive cells is shown, mean ± SEM). Readings were performed 10 days after TRPC3 knockdown. n = 3 biologically independent experiments. **d** p53 and p16 protein levels are shown for cells treated as in (**c**). Data are expressed as mean ± SEM (n = 3). Western blot images are representative of three independent experiments. **e** yH2AX staining and fluorescence microscopy of MRC5 cells treated as in (**c**) showing signs of DNA damage after 10 days of TRPC3 knock-down. % of yH2AX positive cells is shown (mean ± SEM, n = 3 biologically independent experiments). ***P < 0.001 (Student's t test, two-sided). **f** Effect of TRPC3 overexpression in OIS. (Top) colony formation assay reported by crystal violet staining and (bottom) SA-β-gal activity revealed with SA-β-gal staining (% of ß-galactosidase positive cells is shown, mean ± SD). Readings were performed from ER:hRAS-MRC5 cells (bearing doxycycline-inducible TRPC3) in which senescence was induced by 4-OHT treatment over 10 days. To overexpress TRPC3, the cells were kept in doxycycline-containing medium for 3 weeks after senescence induction. **g** Cyclin D1, PCNA1 and p16 protein levels are shown for cells treated as in (**f**) and normalised against Calnexin expression used as loading control. Western blot images are representative of three independent experiments. Scale bar = 100 μm.

readmission only in cells with CIS (Fig. 1f and g). However, in OIS, SOCE was found to be decreased (Fig. 1h and i). Moreover, analysis of single cell responses (examples are shown in Supplementary Fig. 2a, b) revealed that senescent cells are prone to cytosolic $Ca^{2+}$ oscillations. Taken together, these results indicate that $IP_3R$-mediated $Ca^{2+}$ release from the ER is increased in senescence, thus confirming results obtained previously in other models[14].

**TRPC3 regulates senescence.** The above results brought up the question about roles of TRPC3 protein in the acquisition of the senescent phenotype. To address it, we downregulated *Trpc3* expression in HPrFs or MRC5 using siRNA or doxycycline-inducible shRNAs (Supplementary Fig. 2C, D). *Trpc3* knockdown (KD) resulted in senescence induction, reported by proliferation arrest (depicted by the cumulative population doubling, crystal violet and EdU staining: Fig. 2a, b), increased cell size and augmented SA-β-gal activity (Fig. 2c and Supplementary 2e), the activation of the P53/P16 pathways (Fig. 2d) and the formation of heterochromatin foci (Fig. 2e). Of note, as shown by Annexin V/7-AAD staining and flow cytometry analysis (Supplementary Fig. 2f), TRPC3 KD does not induce cell death, what otherwise would compromise the assessment of proliferation rate. Furthermore, downregulation of the TRPC3 homologue, TRPC6, failed to induce senescence (Supplementary Fig. 2g, h), thus highlighting the specific role of TRPC3. Conversely, overexpression of TRPC3 by means of a doxycycline-inducible system (Supplementary Fig. 2i) promoted escape from OIS, as reported by colony formation assays and SA-ß-gal staining (Fig. 2f) showing proliferation rescue after 3 weeks in culture. Escape from senescence was also confirmed by WB analysis of proliferation (Cyclin D1/PCNA1) and senescence (p16 expression) markers, respectively (Fig. 2g).

Altogether, these results indicate that TRPC3 is involved in the establishment of the senescent phenotype.

**TRPC3 constitutively inhibits $IP_3R$-mediated $Ca^{2+}$ release.** Alteration of the ER $Ca^{2+}$ homoeostasis observed in senescent cells (Fig. 1f–i and Supplementary Fig. 2a, b), on the one hand, and the induction of senescence by *Trpc3* KD, on the other hand, suggest that TRPC3 may be involved in the regulation of the ER $Ca^{2+}$-release mechanisms.

To check whether TRPC3 downregulation affects ER $Ca^{2+}$ homoeostasis, we used an experimental strategy similar to the one shown in Fig.1f–i, but on shCTL or shTRPC3-expressing MRC5. Results were similar to those obtained in senescent cells: the shTRPC3-expressing cells showed elevated basal cytoplasmic $Ca^{2+}$, increased histamine-induced $Ca^{2+}$ release from the ER and increased capacitative $Ca^{2+}$ entry (Fig. 3a). The latter results strongly suggest that TRPC3 does not contribute directly to the store-operated $Ca^{2+}$ entry (SOCE), at least in this model.

We also observed that TRPC3-KD cells are prone to spontaneous cytosolic $Ca^{2+}$ oscillations (Supplementary Fig. 3a, b).

Considering that TRPC3 downregulation facilitates cytosolic $Ca^{2+}$ events, we hypothesised that its function is to regulate ER $Ca^{2+}$-release channels rather than to act as an ER $Ca^{2+}$-release channel.

It was previously reported that TRPC3 channel activity can be tuned by its interaction with $IP_3Rs$[37,38] that occurs between the calmodulin/$IP_3$ receptor-binding (CIRB) region on the C-terminal fragment of TRPC3 and the N-terminal fragment of the $IP_3R$[39]. Using absolute qPCR quantification, we found that prostate fibroblasts express all three $IP_3R$ isoforms, of which type 3 accounts for almost 80% (Supplementary Fig. 3c). Western blot analysis performed on subcellular fractions showed that TRPC3 is also expressed in the ER membrane (Supplementary Fig. 3d). Moreover, co-immunoprecipitation and western blot analysis revealed that TRPC3 interacts with $IP_3$ receptor type 3 ($IP_3R3$) in fibroblasts (Supplementary Fig. 3e).

To test whether TRPC3 is involved in the regulation of the $IP_3R$-mediated $Ca^{2+}$ release from the ER, we performed confocal cytosolic $Ca^{2+}$ imaging in cells expressing shCTL or shTRPC3 after stimulation with $IP_3$-AM. We found that downregulation of TRPC3 augmented $IP_3$-induced $Ca^{2+}$ oscillations (Fig. 3b). These results confirm our hypothesis that $IP_3R$ activity is regulated by TRPC3 upon establishment of the senescence phenotype.

For further verification, we conducted confocal $Ca^{2+}$ imaging on prostate fibroblasts stably expressing the ER-targeted ratiometric $Ca^{2+}$ indicator GEM-CEPIA1*er* (Supplementary Fig. 4a)[40]. To assess the rate of the ER $Ca^{2+}$ release directly, fibroblasts bathed in an intracellular-like media with $Ca^{2+}$ clamped at 50 nM (note that $Mg^{2+}$ and ATP were omitted to inhibit SERCA-mediated $Ca^{2+}$ re-uptake) were briefly exposed to 40 μM digitonin (to permeabilize the plasma membrane) during x–y time series imaging (Supplementary Fig. 4b). This revealed that the mean rate of $Ca^{2+}$ release was augmented by TRPC3 knockdown, while TRPC3 overexpression significantly reduced it (Fig. 3c, d). To probe for a possible contribution of $IP_3Rs$ to this release we tested the effect of Xestospongin C (XeC, 5 μM), a well-known inhibitor of $IP_3Rs$[41,42]. As shown in Fig. 3e and f, inhibition of $IP_3Rs$ reduces the permeabilization-unmasked $Ca^{2+}$ release from the ER by about 80%. This indicates that even without exogenous $IP_3R$ stimulation, the observed release is mediated mainly via $IP_3Rs$ and thus depends on TRPC3 expression levels.

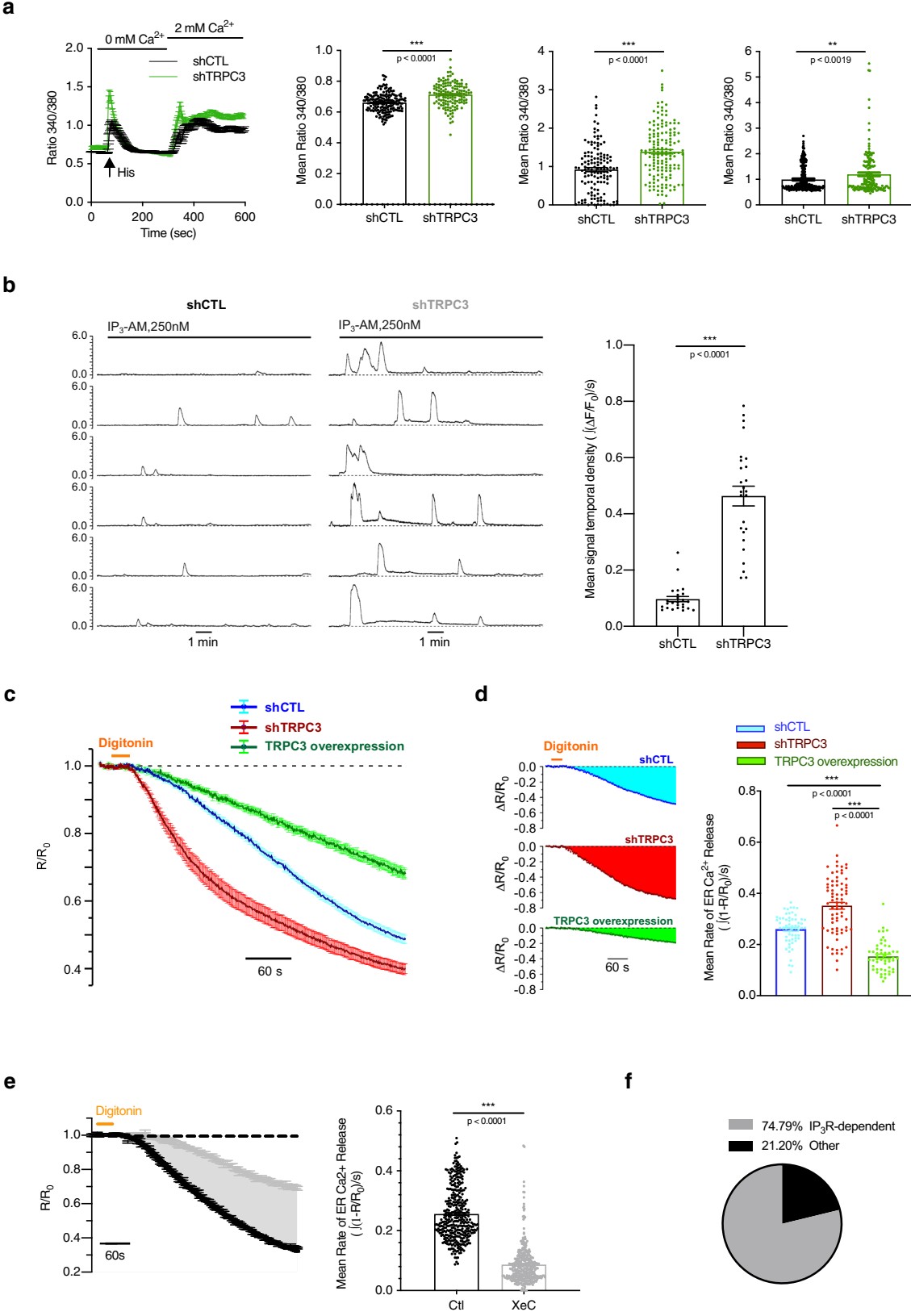

**Fig. 3 TRPC3 inhibits ER Ca$^{2+}$ release. a** Cytosolic Ca$^{2+}$ imaging (Fura-2) performed on HPrFs 10 days after shRNA induction. Left: averaged time courses (mean ± SEM) of fluorescence ratio with excitation wavelengths 340/380 nm. Right: panels comparing the mean values of (1) basal cytosolic Ca$^{2+}$ levels, (2) peak amplitudes of the Histamine (100 μM)-induced ER Ca$^{2+}$ release and (3) peak amplitudes of SOCE, respectively ($n_{shCTL} = 194$, $n_{shTRPC3} = 173$). Data are mean ± SEM ($n = 3$). ***$P < 0.001$ (Student's $t$ test, two-sided). **b** Representative traces (left) of IP$_3$-AM (250 nM)-induced [Ca$^{2+}$] oscillations reported by confocal time-series imaging (at 5 Hz) of Fluo-4 fluorescence in fibroblasts 5 days after shRNA induction: shCTL and shTRPC3, respectively. The bar diagram plot (right) compares corresponding mean signal temporal densities ($n_{shCTL} = 25$; $n_{shTRPC3} = 26$). ***$P < 0.001$ (Student's $t$ test, two-sided). See also Supplementary Figs. 2 and 3. **c** The plots show (mean ± SEM) traces of the self-normalised fluorescence ratio ($R/R_0$, where $R = F_{470-500}/F_{>520}$ at any given time while $R_0$ is $R$ averaged before digitonin application) measured in HPrFs 5 days after shRNA induction ($n_{shCTL} = 60$) ($n_{shTRPC3} = 76$) or overexpressing TRPC3 ($n_{over} = 52$), as indicated. **d** The bar diagram plot compares mean rates of Ca$^{2+}$ release from the ER, calculated as signal mass per second in fibroblasts 5 days after shRNA induction (shCTL, blue), (shTRPC3, red) or overexpressing TRPC3 (green). ***$P < 0.001$ relative to shCTL (Student's $t$ test, two-sided). **e** The same as in (**c**) but measured in wild-type fibroblasts pre-treated with either XeC (grey; $n_{XeC} = 132$) or vehicle only (black; $n_{Ctl} = 107$). The bar diagram plot compares (XeC vs. Ctl) mean rates of Ca$^{2+}$ release from the ER, calculated as signal mass per second. ***$P < 0.001$ (Student's $t$ test, two-sided). **f** The pie chart (right) illustrates IP$_3$R-mediated Ca$^{2+}$ release as a percentage of the total ER release in non-stimulated cells.

To analyse the dependence of the release rate on the IP$_3$ concentration ([IP$_3$]) following cell membrane permeabilization with digitonin, the cells were superfused with a solution supplemented with IP$_3$ at different concentrations ranging from: 0 to 10,000 nM (Supplementary Fig. 4c). The kinetics of Ca$^{2+}$ release from the ER observed at different [IP$_3$] was then compared between MRC5 cells with downregulated (Fig. 4a) or overexpressed TRPC3 (Fig. 4b). This revealed that TRPC3: (1) exerts an inhibitory effect on the Ca$^{2+}$ release at [IP$_3$] = 0–1000 nM (Fig. 4c, d) and (2) reduces the sensitivity of IP$_3$Rs to IP$_3$ at [IP$_3$] < 100 nM (Fig. 4e).

To confirm these findings, we assessed the regulation of IP$_3$R activity by TRPC3 at single channel level using electrical recordings from organelle membrane-derived patches[43]. This approach utilises the incorporation of extracted intracellular membrane fractions into giant uni-lamellar vesicles and allows characterising the activity of the channels from the membranes of cellular organelles in endogenous protein environment and their regulation by partner proteins using standard patch-clamp recording. We found that the open probability of the IP$_3$R channels was markedly reduced in giant uni-lamellar vesicles (GUVs) obtained from TRPC3-overexpressing membranes (Fig. 4f). These results, together with our previous findings (see Supplementary Fig. 3d), suggest that TRPC3 inhibits the IP$_3$Rs from within the ER.

**TRPC3 exerts its effect on IP$_3$R via a direct interaction rather than via its channel activity.** It is well known that IP$_3$R channel activity is sensitive to cytosolic and intraluminal Ca$^{2+}$ concentrations, and is regulated by several partner proteins[22]. To determine whether inhibition of the IP$_3$R-mediated Ca$^{2+}$ release by TRPC3 results from a Ca$^{2+}$-mediated interplay between these two channels or is governed by protein–protein interaction, we transduced MRC5 fibroblasts with the TRPC3 CIRB domain mutant (MKR/AAA) which disrupts the interaction between the two proteins[44]. The expression of this mutant, in contrast to the expression of the wild-type TRPC3, had virtually no effect on the IP$_3$-induced Ca$^{2+}$-release (Fig. 4g and Supplementary Fig. 4d). In addition, the inhibitory effect of TRPC3 overexpression on the IP$_3$R-mediated Ca$^{2+}$ release was only slightly reduced when the pore channel mutants, E630Q ("Ca$^{2+}$ permeation-deficient") or E630K ("pore-dead")[45] were overexpressed instead of wild-type TRPC3 (Fig. 4h and Supplementary Fig. 4e), indicating that TRPC3 exerts its inhibitory effects via direct interaction with IP$_3$R.

**Relief of the TRPC3-mediated inhibition of IP$_3$Rs promotes ER–mitochondria Ca$^{2+}$ transfer and increased OXPHOS facilitating senescence.** Bearing in mind the inhibition of

IP$_3$R-mediated Ca$^{2+}$ release by TRPC3, we hypothesised that the augmented ER Ca$^{2+}$ release caused by TRPC3 downregulation in senescence may lead to enhanced mitochondrial Ca$^{2+}$ uptake, which was previously implicated in senescence induction[18].

To probe mitochondrial Ca$^{2+}$, we transduced MRC5 fibroblasts with the mitochondria-targeted ratiometric Ca$^{2+}$ sensor GEM-GECO1mito[46] (Supplementary Fig. 5a, b) and assessed the effect of TRPC3 downregulation on the mitochondrial Ca$^{2+}$ concentration ([Ca$^{2+}$]$_{mito}$). Comparison of the mean values of GEM-GECO1mito fluorescence ratio at the two wavelength bands in a number of $x$–$y$ confocal images revealed that TRPC3 knockdown resulted in significant elevation of [Ca$^{2+}$]$_{mito}$ (Fig. 5a). Conversely, the elevation of [Ca$^{2+}$]$_{mito}$, observed in OIS, was prevented by the rescue of TRPC3 expression (Fig. 5b) to the same extent as by MCU silencing (Supplementary Fig. 5c, d). Of note, TRPC3 overexpression decreased the basal [Ca$^{2+}$]$_{mito}$ of non-senescent cells, suggesting that TRPC3 also regulates the ER–mitochondrial Ca$^{2+}$ transfer in normal physiological conditions. To elucidate whether the difference in the apparent steady-state [Ca$^{2+}$]$_{mito}$ is related to an alteration of the mitochondrial Ca$^{2+}$ uptake we have conducted $x$–$y$ time series confocal imaging. This revealed that TRPC3 knockdown promotes mitochondrial Ca$^{2+}$ oscillations (Fig. 6a, b) leading to a dramatic augmentation of the mean signal temporal density (Fig. 6c). The latter explains the elevated apparent steady-state [Ca$^{2+}$]$_{mito}$ observed in Fig. 5a. Dynamic confocal imaging of TMRE fluorescence revealed that TRPC3 knockdown promotes spontaneous transient depolarisation of the mitochondrial membrane potential, $\Delta\varphi_{mito}$ (Fig. 6d–f), known as mitochondrial flickers[47].

Taking into account that Ca$^{2+}$ is an important regulator of mitochondrial function[14–16] which is altered in senescent cells[11,48,49], we assessed whether TRPC3 downregulation affects mitochondrial metabolism.

Assessment of the total NAD$^+$/NADH and ADP/ATP ratios demonstrated that TRPC3 knockdown effectively alters cellular metabolism (Supplementary Fig. 6a, b), leading to significant accumulation of NAD$^+$ and elevated total ADP/ATP ratio in siTRPC3-expressing cells. However, in the mitochondrial compartment, TRPC3 downregulation increased ATP levels (Fig. 7a).

We therefore investigated further these alterations by measuring mitochondrial respiration. As expected, TRPC3 downregulation significantly increased mitochondrial oxygen consumption rate (OCR) (Fig. 7b, c). Moreover oligomycin-sensitive respiration, known as ATP turnover, was found to be increased in TRPC3-KD cells relative to siCTL (Fig. 7d), thus, suggesting elevated ATP production. Of note, the enhancement of mitochondrial metabolism observed in TRPC3-KD cells was fully reversed by IP3R3 downregulation (Fig. 7c, d). Conversely,

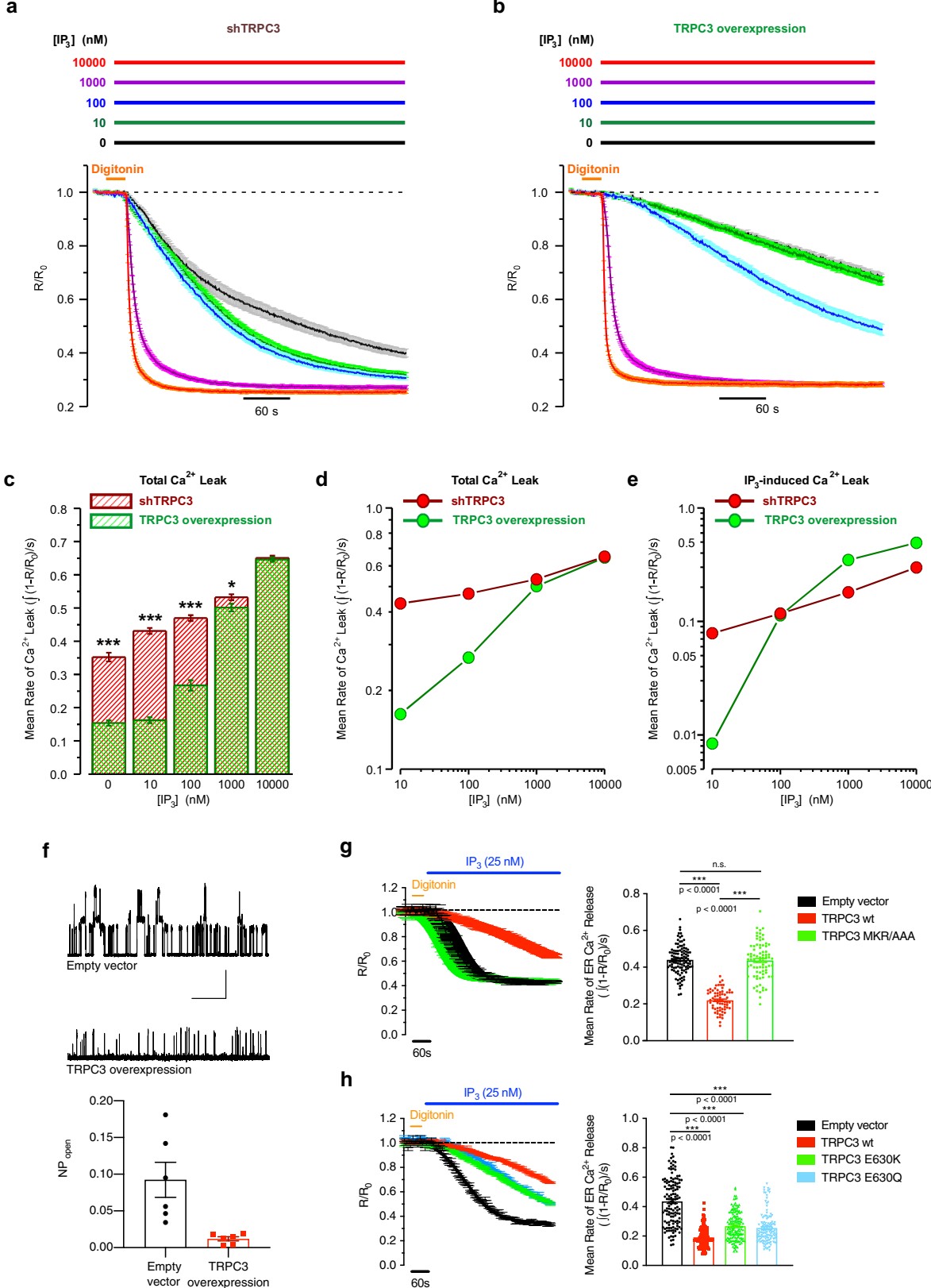

the increased basal OCR and proton leak observed in OIS was prevented by rescue of TRPC3 expression (Supplementary Fig. 7a–e), what reflects the importance of TRPC3 in the regulation of mitochondrial metabolism in senescence. However, the increase of maximal OCR observed in OIS was not abolished by TRPC3 downregulation, what, likely, reflects differences in

regulation of FCCP-stimulated and basal respiration[50]. Monitoring of ROS production (with MitoSox Red) by mitochondria visualised with MitoTracker Deep Red, showed that TRPC3 kD leads to augmented genesis of ROS, which was notably prevented by the concurrent suppression of IP$_3$R3 (Fig. 7e and Supplementary Fig. 6d).

**Fig. 4 TRPC3 reduces the rate of IP$_3$-induced Ca$^{2+}$ release from the ER. a**, **b** Mean traces of the self-normalised GEM-CEPIA1$_{er}$ fluorescence ratio ($R/R_0$) measured at different concentration of IP$_3$ (as depicted above plots in corresponding colours) 5 days after shRNA induction (**b**; $n = 76$, 80, 101, 117 and 80, respectively, for [IP$_3$] = 0, 10, 100, 1000 and 10000 nM) or 3 days after TRPC3 overexpression (**c**; $n = 52$, 113, 51, 104, and 59, respectively for [IP$_3$] = 0, 10, 100, 1000 and 10,000 nM). **c** Bar diagram plot of mean rates of Ca$^{2+}$ release from the ER (derived from data in panels **a**, **b**), calculated as signal mass per second at different concentrations of IP$_3$, after induction of shRNA (red) or TRPC3 overexpression (green) as described above. ***$P < 0.001$, *$P < 0.05$ (Student's $t$ test, two-sided). **d** and **e** Dependence of the mean rate of Ca$^{2+}$ release on [IP$_3$] plotted on logarithmic scale for total (**d**) and IP$_3$-dependent Ca$^{2+}$ release (**e**). Note that the expression of TRPC3 suppresses IP$_3$-mediated Ca$^{2+}$ release at concentrations of IP$_3 < 100$ nM. **f** Left: sample traces of IP$_3$R activity obtained using organelle membrane-derived patch clamp from the ER fractions extracted from cells either transduced with an empty vector (top, $n = 9$) or TRPC3 (bottom, $n = 7$). Right: average NP$_{open}$ summarised in the barplot. *$P < 0.05$ (Student's $t$ test, two-sided). **g** Plots (left) relate traces (mean ± SEM) of the IP$_3$-induced responses (self-normalised GEM-CEPIA1$_{er}$ fluorescence ratio; $R/R_0$) obtained in fibroblasts transduced with empty vector ($n = 108$) to those transduced with wtTRPC3 ($n = 168$) or the TRPC3 CIRB mutant MKR/AAA ($n = 179$) and corresponding bar diagram plots (right) comparing mean rates of Ca$^{2+}$ release. **h** The same as in (**g**) but illustrating the effect of the transduction with TRPC3 pore mutants E630K ($n = 153$) or E630Q ($n = 124$). ***$P < 0.001$ for wtTRPC3 ($n = 156$) vs. empty vector ($n = 131$), #$P < 0.001$ for mutants vs. wtTRPC3 (one-way ANOVA, Tukey's multiple comparisons test). See also Supplementary Fig. 4.

Our flow cytometry analyses revealed that the percentage of senescent cells observed after TRPC3 knockdown was reduced following: (1) inhibition of Ca$^{2+}$ transfer to mitochondria by Ru360, (2) IP$_3$R3 knockdown or (3) mitochondrial ROS scavenging by MitoTempo (Supplementary Fig. 6h). Altogether, these observations unravel a chain of signalling events caused by TRPC3 downregulation and leading to enhanced OXPHOS and elevated mitochondrial ROS production (reported to maintain senescence[18]), providing an additional evidence for the role of TRPC3 in modulation of the ER–mitochondrial Ca$^{2+}$ transfer and mitochondrial metabolism observed in senescence.

**TRPC3 downregulation is characterised by a tumour-promoting SASP**. The SASP secretion supports tumour progression, since it promotes cancer cell proliferation, resistance to chemotherapy and a pro-inflammatory microenvironment[4]. In this context, mitochondrial metabolism seems to play a central role[12]. However, mitochondrial dysfunction has also been associated with another secretory phenotype, lacking the IL-1-dependent inflammatory arm and altering the differentiation of preadipocytes and keratinocytes[11]. To discriminate between these two distinct phenotypes, we analysed the SASP secreted upon TRPC3 downregulation. Antibody arrays covering a large spectrum of cytokines, chemokines and growth factors, showed that TRPC3 downregulation is associated with a pro-inflammatory and pro-tumour SASP, notably including IL-8, ENA-78, GRO-alpha and other soluble factors associated with tumour progression and aggressiveness (Fig. 8a and Supplementary Fig. 8a).

Because TRPC3 KD cells show an important secretion of SASP per se, we next aimed to check if TRPC3 overexpression in senescent cells would modify their SASP expression. To this aim, we rescued TRPC3 in OIS and checked, by qRT-PCR, the expression of some of the main SASP components. Our data confirm further the role of TRPC3 in the regulation of the senescent cells secretome, since its overexpression led to a drastic reduction of SASP (Supplementary Fig. 8b). To test whether this SASP exerted any pro-tumour effect in vitro, we incubated PC3 and DU145 prostate epithelial cancer cells with a medium conditioned with prostate cancer associated fibroblasts (CAFs) expressing either shTRPC3 or shCTL. Comparison of the effect of the conditioned medium (CM) from the two groups of CAFs on the proliferation rate of the epithelial cells revealed only modest effect of the shTRPC3: ~35% increase in the epithelial cell proliferation rate (Fig. 8b). Of note, the effects of conditioned medium from OIS cells on prostate cancer cell proliferation were almost completely abolished by rescue of TRPC3 expression (Fig. 8c).

To verify whether modification of the SASP caused by TRPC3 downregulation could contribute to the acquisition of a chemotherapy-resistant phenotype, we assessed the effect of SASP secreted by CAFs on prostate cancer cells. To this end, we cultured CAFs expressing either shCTL or shTRPC3 for 24 h and then exposed PC3 and DU145 prostate cancer cells to the corresponding conditioned medium. Prostate cancer cells treated with non-conditioned medium were used as control. Following a 24 h incubation, the cells were transferred to media containing either docetaxel or vehicle (DMSO) only and cultured for an extra 48 h. Cell viability was then measured for all six conditions. The chemotherapy-resistance was expressed as a ratio of the cell viability observed after treatment with docetaxel to that observed with vehicle only (Fig. 8d). This approach revealed that in both prostate cancer cell models tested, the resistance to docetaxel was significantly stronger following incubation with conditioned medium derived from the CAFs expressing shTRPC3: about 25% increase in cell viability relative to that observed with the medium derived from shCTL-expressing CAFs. Of note, the incubation in the conditioned medium derived from shCTL-expressing CAFs, per se increased the resistance to DOX, but only in PC3 cells. Thus, our results reveal that TRPC3 suppression in both normal and cancer-associated fibroblasts modify the behaviour of epithelial cancer cells. To assess further our hypothesis that TRPC3 suppression in stromal cells favours a senescent pro-tumour microenvironment, we have conducted in vivo experiments. In these experiments, HPrFs carrying doxycycline-inducible shTRPC3 or shCTL were pre-treated with doxycycline for 7 days to induce senescence. These cells were then mixed with PC3-Luc cells in proportion 1:2 and injected subcutaneously into immunodeficient mice. The animals were then imaged once a week during 6 weeks. The results of the bioluminescence imaging and tumour size measurements revealed that TRPC3 downregulation in prostate stromal cells accelerates growth of the newly formed subcutaneous prostate tumours (Fig. 8e–g). This suggests that stromal cells senescence caused by TRPC3 downregulation facilitates cancer progression mainly via the production of a pro-inflammatory and pro-tumour SASP.

## Discussion

Our study uncovers a molecular mechanism underlying mitochondrial dysfunction that contributes to pro-tumour effects of senescence. In particular, we demonstrated that, in proliferating cells, TRPC3 inhibits the IP$_3$R–mediated Ca$^{2+}$ release thus limiting the ER–mitochondria Ca$^{2+}$ transfer, while the development of senescent phenotype is associated with TRPC3 downregulation. This, in turn, results in augmented mitochondrial Ca$^{2+}$ uptake/load, mitochondrial depolarisation, elevated ROS production and mitochondrial metabolism alteration.

**a**

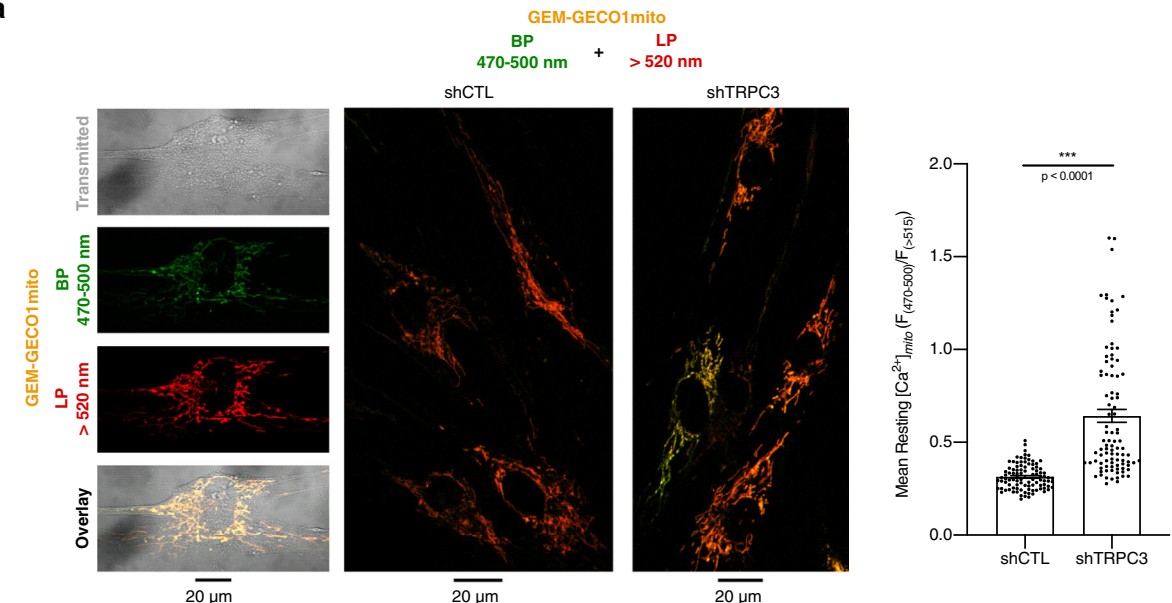

**b**

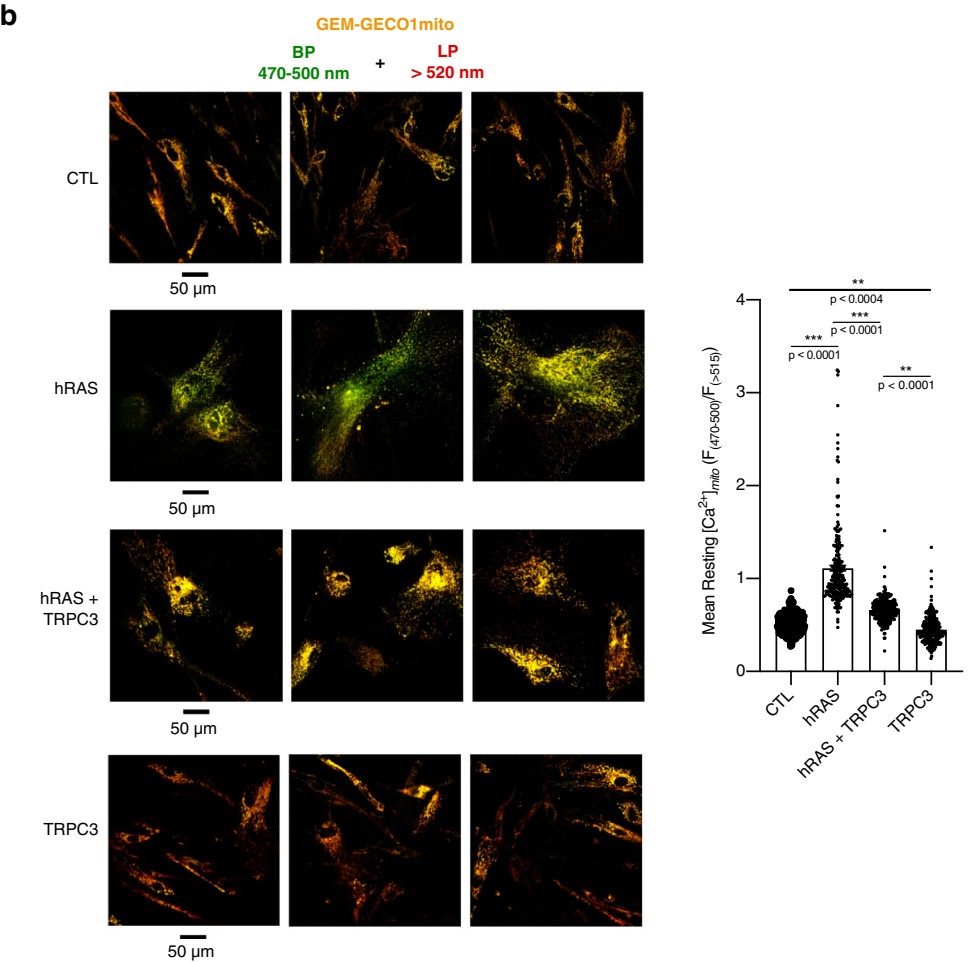

**Fig. 5 The level of TRPC3 in fibroblasts affects basal mitochondrial Ca$^{2+}$ load. a** The gallery (left) shows (from top to bottom) the transmitted light image of a single fibroblast, confocal image of GEM-GECO1$_{mito}$ fluorescence captured at wavelengths 470–500 nm, at wavelengths >520 nm and their overlay. The gallery (middle) compares confocal images (overlay of 2 channels: BP 470–500 nm plus LP > 520 nm) of GEM-GECO1$_{mito}$ fluorescence from the fibroblasts transduced with shCTL and shTRPC3, as indicated. The bar diagram plot (right) compares mean resting [Ca$^{2+}$]$_{mito}$ (estimated as $F_{470-500}$/$F_{>520}$) in the fibroblasts transduced with shCTL ($n = 96$) and shTRPC3 ($n = 96$). ***$P < 0.001$ relative to shCTL (Student's $t$ test, two-sided). **b** Elevation of mitochondrial Ca$^{2+}$ load upon hRAS overexpression is markedly prevented by simultaneous overexpression of TRPC3. The gallery (left) compares confocal images (overlay of 2 channels: BP 470–500 nm plus LP > 520 nm) of GEM-GECO1$mito$ fluorescence from control MRC5, MRC5 with hRAS-induced senescence (hRAS) and MRC5 overexpressing hRAS and TRPC3 simultaneously (hRAS + TRPC3) or TRPC3 only (TRPC3). Images of three representative fields of view per condition are shown. The bar diagram plot (right) compares mean resting [Ca$^{2+}$]$_{mito}$ (estimated as $F_{470-500}$/$F_{>520}$) in control MRC5 (Control: $n = 390$) with that in MRC5 with hRAS-induced senescence (+hRAS: $n = 232$) and MRC5 overexpressing hRAS and TRPC3 simultaneously (+hRAS + TRPC3: $n = 266$) or TRPC3 only (+TRPC3: $n = 251$). ***$P < 0.001$ relative to control, unless stated otherwise. One-way ANOVA, Tukey's multiple comparisons test.

This inhibition of IP$_3$Rs by TRPC3 appeared to be unrelated to its potential activity as Ca$^{2+}$-permeable channel. Indeed, on the one hand, downregulation of TRPC3 was associated with significant increase in the ER Ca$^{2+}$ release, while its overexpression had the opposite effect (Fig. 3c, d). The observed effects cannot be attributed to a potential variability in cytosolic Ca$^{2+}$ concentration ([Ca$^{2+}$]$_c$) or SERCA activity, since the measurements were performed on permeabilized cells superfused with Mg$^{2+}$/ATP-free solution with [Ca$^{2+}$] clamped at 50 nM. On the other hand, the observed increase of the mitochondrial Ca$^{2+}$ uptake/load upon TRPC3 downregulation (Figs. 5 and 6) rules out that TRPC3-mediated mitochondrial Ca$^{2+}$-uptake is the main mechanism controlling ER Ca$^{2+}$ homoeostasis. All the above suggest that the most likely mechanism, which determines the dependence of the ER Ca$^{2+}$ release on TRPC3 bioavailability, is the regulation of other Ca$^{2+}$-permeable channel(s) in the ER membrane by TRPC3. IP$_3$R is commonly recognised as the major conduit contributing to the ER–mitochondria Ca$^{2+}$ transfer[51]. Inhibition of the ER Ca$^{2+}$ release by the IP$_3$R inhibitor XeC[42] indicates that IP$_3$Rs are responsible for about 80% of the Ca$^{2+}$ release from the ER of prostate stromal cells (Fig. 3e, f). Our co-immunoprecipitation experiments confirmed the interaction between TRPC3 and IP$_3$R3 proteins (Supplementary Fig. 3e), while the dynamic confocal imaging of the [Ca$^{2+}$]$_{ER}$ changes in permeabilized stromal cells (Fig. 4a–f) and patch-clamp recording of the IP$_3$R single channel activity (Fig. 4g) provided the most direct evidence of IP$_3$R activity inhibition by the TRPC3 protein. The fact that this inhibition is caused by the TRPC3–IP$_3$R interaction rather than the TRPC3 channel activity was confirmed further with TRPC3 mutants (Fig. 4h, i).

The ER–mitochondria Ca$^{2+}$ transfer is essential for cell metabolism, since mitochondrial Ca$^{2+}$ uptake is crucial for the regulation of the oxidative phosphorylation and ATP production. On the one hand, in many cell types, reduction of the constitutive IP$_3$R-mediated Ca$^{2+}$ transfer to mitochondria results in autophagy that opposes a bioenergetic crisis and promotes cell survival[17,52]. However, in cancer cells, the induction of autophagy is not sufficient for cell survival, since necrotic cell death prevails due to mitochondrial dysfunction[53]. In this context, the authors showed that suppression of about 80% of the ER-Ca$^{2+}$ release events (and the consequent loss of Ca$^{2+}$ transfer to the mitochondria) following xestospongin application compromised cell bioenergetics and evoked either pro-survival macroautophagy in normal cells, or cell death in transformed or tumour cells. However, the extent of mitochondrial Ca$^{2+}$ uptake alteration in these cells was not reported[17,53]. On the other hand, a sustained increase of mitochondrial Ca$^{2+}$ concentration induces apoptosis, since mitochondrial Ca$^{2+}$ overload leads to mitochondrial swelling and, hence, to perturbation/rupture of the outer mitochondrial membrane, release of mitochondrial apoptotic factors into the cytosol and activation of the caspase cascade[54].

Another possible consequence of mitochondrial dysfunction is the induction of senescence. Indeed, several papers associate ROS production with the induction and stabilisation of senescence, as well as with OXPHOS[55]. Our results confirm further the role of mitochondrial homoeostasis and bioenergetics in senescence, since TRPC3 downregulation leads to increased OXPHOS (reported by augmented NAD+/NADH and ADP/ATP levels) and augmented mitochondrial ROS production.

The spatio-temporal pattern of Ca$^{2+}$ signalling is an important determinant of cell fate[56]. One of the major findings of our paper is that the relief of the TRPC3-mediated IP$_3$R inhibition caused by TRPC3 downregulation facilitates [Ca$^{2+}$] oscillations in the cytosol and mitochondria rather than sustained elevation of [Ca$^{2+}$]$_{mito}$. This specific oscillatory Ca$^{2+}$ signalling in prostate stromal cells seems then to be a crucial point for the choice of the induction of senescence as an alternative to autophagy or apoptosis. Although senescent cells are more prone to Ca$^{2+}$ oscillations than their proliferative counterpart, we observed that SOCE amplitude in senescence may vary. This is not surprising, since it is generally recognised that different Ca$^{2+}$ signatures can lead to a same cell fate[33]. However, understanding and deciphering these differences is a challenge that deserves to be explored to better characterise the role of Ca$^{2+}$ signalling in senescence.

Senescence in the microenvironment exerts a well-documented pro-tumour effect. Indeed, SASP content is enriched in soluble factors which facilitate (1) inflammation-related cancer progression and immune escape and (2) cancer cell proliferation and resistance to anticancer therapies. Our results demonstrate that senescence is associated with mitochondrial dysfunction depending on TRPC3 bioavailability and is characterised by the release of pro-inflammatory chemokines, cytokines and other tumour-promoting molecules (Fig. 8a–c). Besides the pro-inflammatory chemokines and cytokines evoking the induction of the IL-1-regulated arm of SASP, some of the soluble factors released by fibroblasts upon TRPC3 downregulation such as IL-8, HGF, ENA-78, CXCL5, RANTES, PDGF Rb and MMP1 directly stimulate tumour growth and resistance to chemotherapy or contribute to tumour progression in vivo (Fig. 8b-g)[57–60]. However, the discrepancies observed between the rate of cell proliferation in vitro and tumour growth in vivo strongly suggest the contribution of some other factors to the latter.

Altogether, our findings pinpoint TRPC3 as a key regulator of intracellular Ca$^{2+}$ events engaged in senescence affecting SASP production and thereby tumour progression.

## Methods

**Cell lines and primary cultures.** Human primary lung embryonic fibroblasts MRC5 (PD 5-40, ATCC, USA) and Virus-packaging GP293 cells (Clontech, USA)

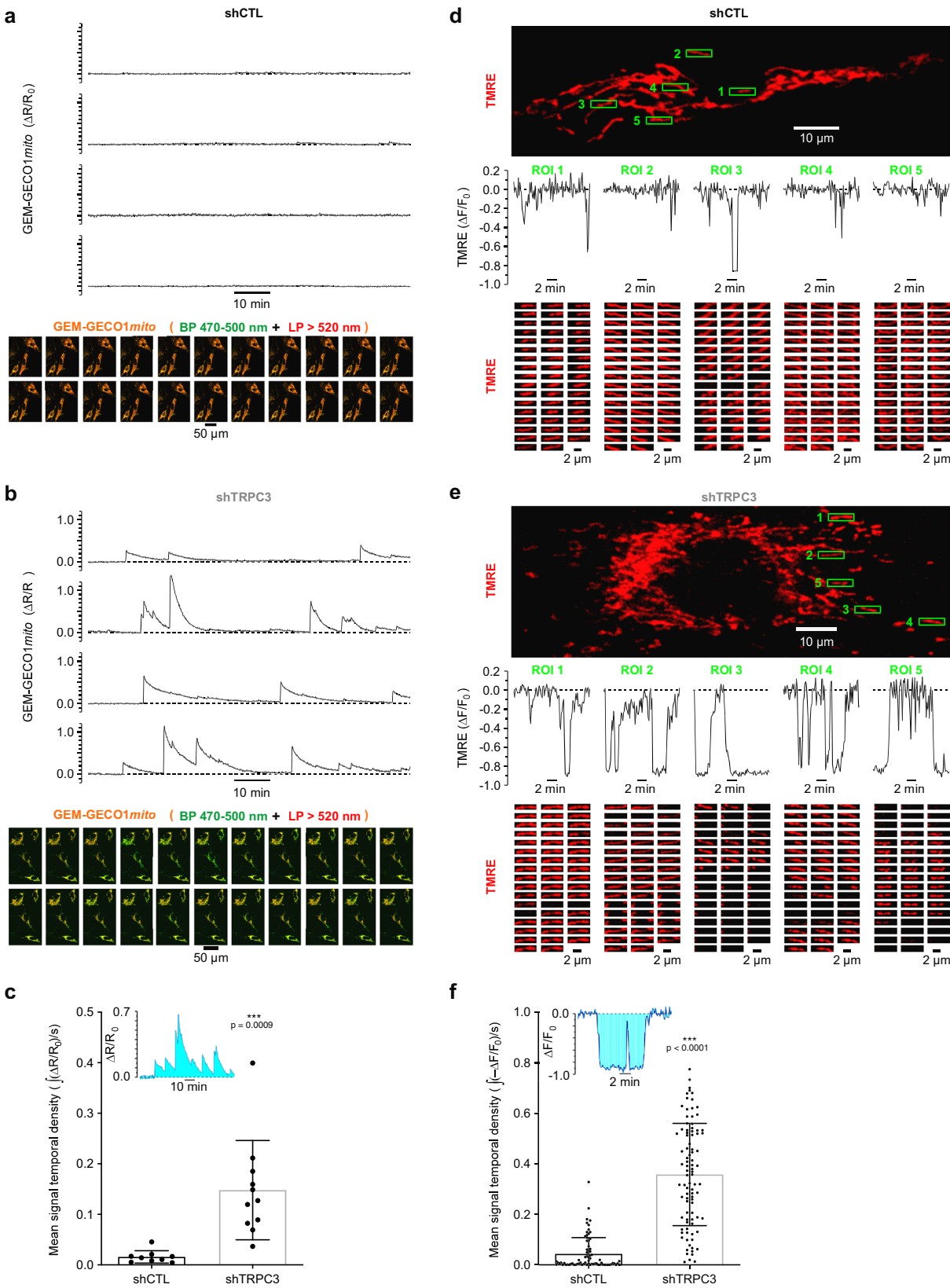

were cultured in Dulbecco's modified Eagle's medium (DMEM) supplemented with 10% foetal bovine serum and penicillin/streptomycin 100 U ml⁻¹. Human primary prostate fibroblasts (PD 0-15, ScienCell, USA) were cultured in MCDB131 medium (Biological Industries, Israel) supplemented with 10% foetal bovine serum, 10 mM

Hepes, 1 ng ml⁻¹ bFGF and 0.1 ng ml⁻¹ EGF. CAFs from human prostate cancer biopsies were obtained by explant as described in our previous paper[61].

PC3, PC3-Luc and DU145 epithelial prostate cancer cells (ATCC) were cultured in RPMI 1640 supplemented with 10% FBS and 1% L-glutamine. HEK293T

**Fig. 6 Silencing of TRPC3 promotes spontaneous Ca²⁺ oscillations and transient membrane depolarisations in mitochondria. a–c** Ratiometric confocal $[Ca^{2+}]_{mito}$ imaging (at 0.2 Hz) was performed in fibroblasts expressing GEM-GECO1*mito* and transduced with either shCTL (**a**) or shTRPC3 (**b**). The plots (top) show temporal profiles of the GEM-GECO1*mito* self-normalised fluorescence ratio ($\Delta R/R_0$; where $R = F_{470-500}/F_{>520}$ at any given time, while $R_0$ is a mean of at least 50 sample minimums of $R$) for four cells. The galleries (bottom) show every 50th image (overlay of 2 channels: BP 470–500 nm plus LP > 520 nm) captured during entire observation period (traces: top). **c** The bar diagram plot compares corresponding mean signal temporal densities, calculated as signal mass ($\int(\Delta R/R_0)$) per second, in shCTL ($n = 9$) and shTRPC3 fibroblasts ($n = 11$). ***$P < 0.001$, Student's $t$ test, two-sided. **d–f** Changes in the mitochondrial membrane potential (at 0.1 Hz) were monitored in TMRE-loaded fibroblasts transduced with either shCTL (**d**) or shTRPC3 (**e**). The plots (middle) show temporal profiles of the TMRE self-normalised fluorescence ratio ($\Delta F/F_0$) at 5 boxed (top) regions of interest (ROIs), as indicated. The galleries (bottom) show every 2nd image (left to right, top to bottom) captured at corresponding ROIs. **f** The bar diagram plot compares corresponding mean signal temporal densities in shCTL ($n = 6$; 75 ROIs) and shTRPC3 ($n = 5$; 92 ROIs) fibroblasts. ***$P < 0.001$ (Student's $t$ test, two-sided).

(Dharmacon) and GP293 packaging cells were cultured in DMEM containing 10% FBS. The cells were maintained at 37 °C under a 10% $CO_2$ atmosphere.

Sf-9 insect cells (Novagen) were cultured in Grace's Insect Medium, supplemented with 5% FBS at 28 °C in a ventilated incubator. All the media and supplements, if not stated otherwise, were from Gibco (CA, USA).

**Vectors and plasmid construction.** The plasmid pNLCX2-hRAS^G12V^:ER, a gift from Masashi Narita (Addgene plasmid # 67844) was used to transfer the mutated RAS oncogene in MRC5 cells. The endoplasmic reticulum Ca²⁺ fluorescent genetic reporter "CEPIA1-er", a gift from Masamitsu Iino (Addgene plasmid # 58217), was PCR modified to insert the StuI restriction site and then cloned into the pLNCX2 retroviral vector (Clontech) by HindIII/StuI. The mitochondria Ca²⁺ fluorescent genetic reporter "CMV-mito-GEM-GECO1", a gift from Robert Campbell (Addgene plasmid # 32461), was cloned into the pLNCX2 vector by BamHI-BglII/HindIII.

EGFP-TRPC3 was produced by E-Zyvec (France) and cloned into the pLNCX2 vector by SalI/ClaI.

Wild-type "TRPC3-Myc", a gift from Craig Montell (Addgene plasmid # 25902), CIRB mutants "TRPC3-MRK/AAA"[44] and the pore mutants "TRPC3 E630K-E630Q" were cloned into the baculoviral vector pFastBac LIC, a gift from Scott Gradia (Addgene plasmid # 30111), previously modified by LIC cloning to insert the CMV promoter (pFastBac-CMV).

**Virus production, gene transfer and gene silencing.** To produce retroviral particles, GP-293 packaging cells were transfected with the corresponding pLNCX2 plasmid and the pVSV-g vector (Clontech) using PEIpro Transfection reagent (PolyPlus, France). After 48 h, viral supernatant was then filtered and used to transduce target cells. Bacmids, obtained after transformation of DH10Bac bacteria (Life Technologies), were transfected into Sf9 cells (Novagen) by the Cellfectin-II reagent (ThermoFisher) to produce P1 baculoviral stock. P2 stock, obtained by infecting fresh Sf9 cells, was then titrated by qPCR and target cells were transduced at MOI = 500.

Lentiviruses encoding shRNA, siRNAs and controls were purchased from Dharmacon (Lafayette, USA). Target cells were transduced at MOI = 1.

Cells stably expressing the construct were selected by G418 (400 µg/ml) or puromycin (1 µg/ml) for 7 days.

**Conditioned medium experiments.** Conditioned medium was obtained by incubating HPrFs, previously treated with doxycycline for 1 week to induce TRPC3-mediated senescence, for 24 h in RPMI 1640 medium 0% FBS. PC3-Luc or DU145 cells were plated into 96-well plates ($4 \times 10^3$) and incubated for a further 72 h with the conditioned medium. Cell viability/proliferation was calculated by MTS assay.

**Cell proliferation and viability.** Cell proliferation and viability were assessed either by haemocytometer counting or trypan blue exclusion assay. MTS or bioluminescence assays were performed according manufacturer's protocols and plates were read at 490 or 530 nm with TriStar² LB 942 microplate reader (Berthold, Germany).

To calculate cumulative population doublings, the following formula was applied: PDL = 3.32 ($\log X_e - \log X_b$) + S, where $X_b$ = cell number at the beginning of the incubation time, $X_e$ = the cell number at the end of the incubation time, and S is the starting PDL of the inoculum.

For crystal violet staining, cells were washed twice with PBS, fixed with methanol for 5 min, incubated with 0.5% crystal violet for 30 min and rinsed with distilled water.

**SA-β-Gal analysis and growth assay.** SA-β-Gal activity was determined according to Debacq-Chainiaux F et al.[32] for cytochemistry and flow cytometry, except that C12FDG was replaced by 10 µM DDAOG (Gong et al., 2009).

**Quantitative real-time PCR.** Total RNA was prepared using the NucleoSpin® RNA Plus (Macherey Nagel). Two µg of RNA were reverse-transcribed using random hexamers, MuLV and dNTPs (Life Technologies) in a final volume of 20 µl according to manufacturer's instructions. Quantitative Real-time PCR reactions were performed using the CFX96 Real-time PCR system (Biorad). Primers were designed with the qPrimerDepot software (http://primerdepot.nci.nih.gov/) and are listed in Supplementary Table 2. For primers used to amplify TRP transcripts, see the Declaration of Interests. PCR products were measured by EVA Green fluorescence (SsoFast™ EvaGreen® Supermix, Biorad). Experiments were performed in triplicates for each data point. Results were analysed with the CFX Manager version 3.1 software (BioRad). Gene expression was normalised to GAPDH, ACTIN or 18S gene, and fold expression relative to the control is shown.

For absolute qPCR, plasmids containing the sequences for human IP₃R1, IP₃R2 and IP₃R3 and for GP64 viral glycoprotein were serially diluted to obtain a calibration curve and used in a standard reaction.

**Co-immunoprecipitation, immunoblotting and antibody arrays.** For western blots, cells lysis was performed with different solutions depending on the analysed proteins and protocols. The total protein content was measured using the BCA assay. Immunoprecipitation was performed on cell lysates prepared as above and diluted in 1 ml of lysis buffer. Agarose-ProteinA/G beads (SantaCruz) were incubated either with rabbit anti-TRPC3 (5 µg/mg lysate) or rabbit anti-IP₃R3 (5 µg/mg lysate) for 3 h at +4 °C. Proteins were then incubated overnight at +4 °C with the Ab-coupled beads and washed five times in IP buffer followed by western blot.

Proteins were resolved by SDS–PAGE, transferred to PVDF or nitrocellulose membranes (Hybond-C extra, Healthcare Life Sciences, Piscataway, NJ, USA or Immobilion-P membranes, Millipore) and incubated with the following primary antibodies: rabbit polyclonal anti-TRPC3 (1/200) Alomone Labs Cat# ACC-016; mouse monoclonal anti-GAPDH clone 6C5 (1/2000) Abcam Cat# ab8245; mouse monoclonal anti-Beta-Actin clone AC-74 (1/2000) Sigma Aldrich Cat# A2228; mouse monoclonal anti-IP3R3 clone 2/IP3R-3 (1/1000) BDBiosciences Cat# 610313; rabbit monoclonal anti-N-Cadherin clone H-63 (1/200) SantaCruz Biotech. Cat# sc-7939; mouse monoclonal anti-Calnexin clone C8.B6 (1/2000) Millipore Cat# MAB3126; mouse monoclonal anti-VDAC1 clone B-6 (1/200) SantaCruz Biotech. Cat# sc-390996; mouse monoclonal anti-Cytocrome C clone A-8 (1/200) SantaCruz Biotech. Cat# sc-13156; rabbit monoclonal anti-MCU clone D2Z3B (1/1000) Cell Signalling Cat# 149975; mouse monoclonal anti-P53 clone DO-1 (1/200) SantaCruz Biotech. Cat# sc-126; mouse monoclonal anti-PCNA1 clone PC10 (1/200) SantaCruz Biotech. Cat# sc-56; mouse monoclonal anti-Cyclin D1 clone G124-326 (1/1000) BD Pharmingen. Cat# 554180; goat polyclonal anti-p16INK4a/CDKN2A (1 µg/mL) R&D systems Cat# AF5779; rat monoclonal anti-hRAS clone 342404 (5 µg/mL) R&D systems Cat# MAB3429.

Secondary antibodies used were peroxidase-conjugated (Jackson ImmunoResearch Laboratories, West Grove, PA, USA). Peroxidase activity was revealed using an enhanced chemiluminescence (ECL) or ECL advanced kit (GE Healthcare Life Sciences). Band densities were quantified using the ImageJ/Fiji version 1.53 software. Density of each band was divided by the density of the reference protein band or related control band, and the obtained value was normalised with respect to that obtained under control conditions. Full uncropped and unprocessed blots are available in the Source Data file. Secreted proteins were detected using a RayBio® Human Cytokine Array C2000 (RayBioTech). Briefly, conditioned medium was generated from cells (equal number of cells per condition) following a PBS wash and incubation in serum-free RPMI for 48 h. Conditioned medium was then filtered (0.2 µm) and incubated on the array overnight at 4 °C. Washes and detection procedures were performed as indicated by the manufacturer, and chemiluminescence signals were analysed with the ImageJ/Fiji version 1.53 software.

**Ca²⁺ imaging and electrophysiology**

*Cell permeabilization.* Where applicable, cells were permeabilized "online" during imaging protocol by 20 s exposure to 40 µM Digitonin using home-made local perfusion system.

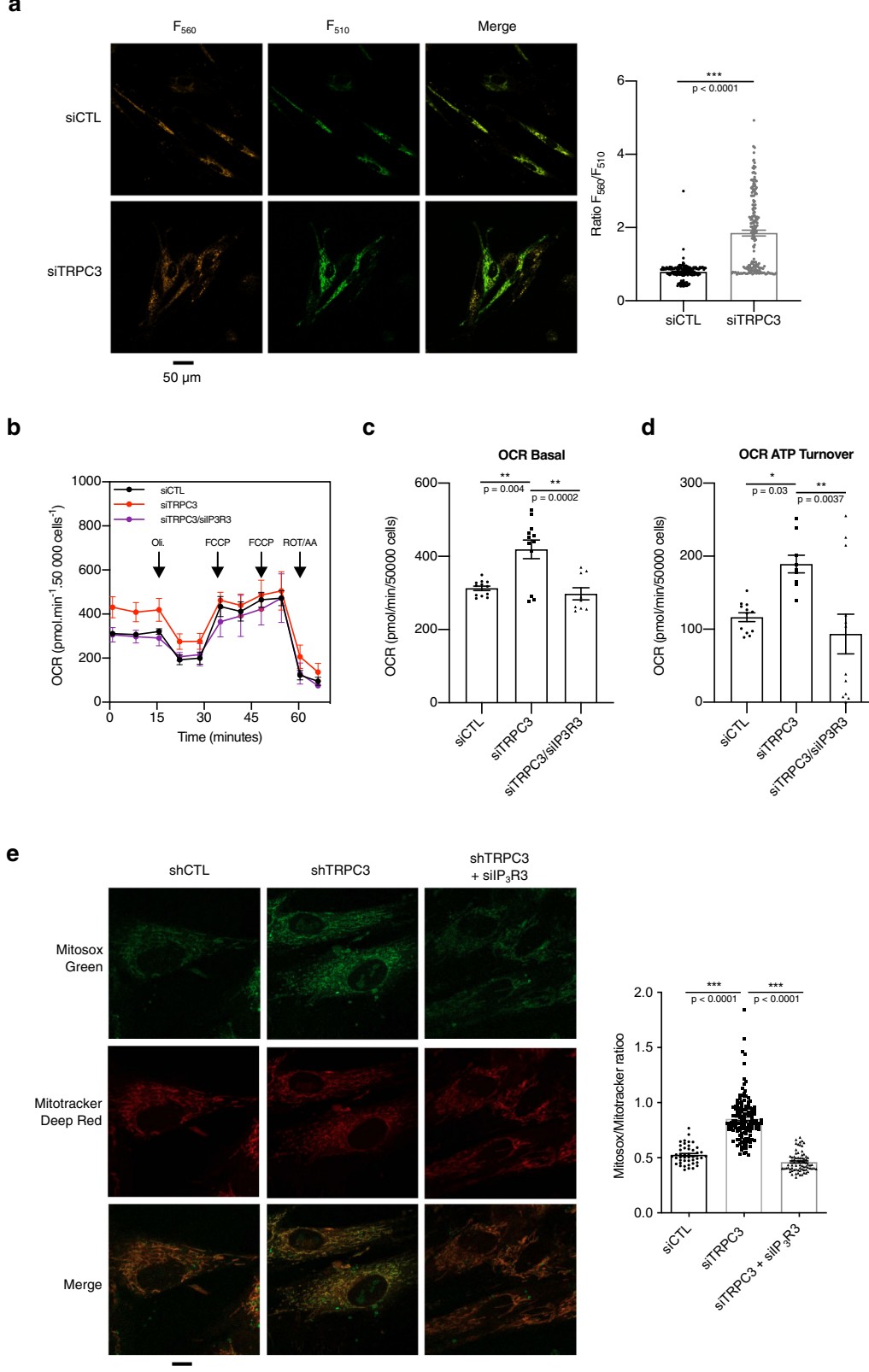

*Confocal microscopy.* Confocal imaging was performed with LSM 510 META confocal workstation using a Plan-Neofluar ×40 1.3 NA objective (Carl Zeiss, Germany).

Fluo-4, Mitotracker Green and eGFP were excited by the 488 nm line of 500 mW Argon ion laser (Laser-Fertigung, Hamburg, Germany) and fluorescence

was captured at wavelengths 505–545 nm. TMRE and ER-Tracker Red were excited by the 543 nm line of 5 mW He/Ne laser (Laser-Fertigung, Hamburg, Germany) and fluorescence was captured at wavelengths >560 nm. CellMask Deep Red was excited by the 633 nm line of 15 mW He/Ne laser (Laser-Fertigung, Hamburg, Germany) and the fluorescence was captured at wavelengths >650 nm.

**Fig. 7 Relief of the TRPC3-mediated inhibition of IP$_3$Rs promotes senescence via facilitation of ER-mitochondria Ca$^{2+}$ transfer, alteration of mitochondrial metabolism and ROS production. a** The galleries (from left to right) show the confocal images of the GO-ATeam probe fluorescence excited at 488 nm and captured at wavelengths of 500–520 nm, >550 nm and their overlay. The bar diagram plot (right) compares mean ATP levels (estimated as $F_{500--520}/F_{>550}$) in the MRC5 transfected with siCTL ($n = 162$) or siTRPC3 ($n = 178$). ***$P < 0.001$ relative to siCTL (Student's $t$ test, two-sided). **b** Oxygen consumption rate (OCR in pmol O$_2$ min$^{-1}$ 50,000 cells$^{-1}$) measured using the Seahorse XFe24 technology (mean ± SD; $n = 4$) in fibroblasts transfected with either siCTL, siTRPC3 or siTRPC3/siIP3R3. Inhibitors have been injected as indicated by arrows: Oligomycin A (Oli.), carbonyl cyanide-4-(trifluoromethoxy)phenylhydrazone (FCCP), and rotenone and antimycin A (Rot/AA). **c** and **d** States of mitochondrial respiration are "basal" (**c**) for basal respiration and "ATP Turnover" (**d**) for ATP-linked OCR (basal measurement minus oligomycin response). Mean ± SD; $n = 4$ (one-way ANOVA followed by Tukeys's post-test *$P < 0.05$). See also Supplementary Fig. 6e–g. **e** Confocal images of Mitosox Green and Mitotracker Deep Red fluorescence (top) and bar diagram plot (bottom) of Mitosox/Mitotracker fluorescence ratio comparing mitochondrial ROS production in fibroblasts transduced either with shCTL ($n = 74$) or shTRPC3 ($n = 143$) and in siIP3R3-transfected fibroblasts transduced with shTRPC3 ($n = 74$). Mean ± SD ***$P < 0.001$, **$P < 0.01$ (one-way ANOVA, Tukey's multiple comparisons test).

To verify optical channel setting for ratiometric confocal [Ca$^{2+}$]$_{ER}$ or [Ca$^{2+}$]$_{mito}$ imaging, fluorescence emission spectra of GEM-CEPIA1er and GEM-GECO1$mito$ expressed in fibroblasts were measured after cell permeabilization (with 40 μM Digitonin) and incubation in the solution with [Ca$^{2+}$] buffered (EGTA) to 50 nM or 1 mM (for GEM-CEPIA1er), or in control and after application of 2.5 μM ionomycin (for GEM-GECO1 $mito$). Fluorescence was excited by 405 nm line of blue diode laser and captured by META unit of LSM 510 META confocal workstation. Upon time-series [Ca$^{2+}$] imaging, fluorescence then was captured at the two channels: green—470–500 nm and red—>520 nm in line-by-line mode of the multitrack configuration of the confocal scanner.

Illumination intensity was attenuated to 0.6–4% (depending on fluorophore used) with an acousto-optical tunable filter.

Image processing was carried out using LSM 5 software (Zeiss, Oberkochen, Germany) and with custom routines written in IDL (Research Systems, Inc., Boulder, CO, USA).

Fura-2 was alternatively excited at 340 and 380 nm with a monochromator (Polychrome IV; TILL Photonics) and was captured after filtration through a long-pass filter (510 nm) by a 5 MHz charge-coupled device camera (MicroMax; Princeton Instruments). Acquisition and analysis were performed with Metafluor 4.5 software (Universal Imaging Corp.).

*Organelle membrane-derived patches*. The collection of the ER-containing fractions from prostate fibroblasts transduced either with empty vector or TRPC3, the preparation of GUVs and the patch-clamping were performed following the protocol described in our previous paper[43].

**Flow cytometry**. Flow cytometry was performed using a CYAN AD (Beckman Coulter, USA) and the Summit Software version 4.3. Briefly, $1 \times 10^6$ cells were incubated with 5 μl of Pacific Blue Annexin-V and 5 μl of 7-AAD for 15 min at +4 °C. Collected data were exported to and analysed with the FlowJo 7.0 software (FlowJo, LLC).

**Mitochondrial membrane potential**. Mitochondrial membrane potential flickers were analysed by utilising the TMRE probe (0.1 μM). Cells were incubated with TMRE for 1 min and analysed by confocal microscopy to determine the number of flickers over a time period of 5 min.

**NAD$^+$/NADH and ADP/ATP ratios**. MRC5 cells were transfected with siTRPC3 or siCTL and cultured for 7 days. Cells were then plated in 96-well plates for measurements. The NAD$^+$/NADH ratio was determined using the NAD/NADH-Glo Assay (Promega, G9071) according to the manufacturer's instructions. The ADP/ATP ratio was determined using an ADP/ATP Ratio Assay kit (Sigma, cat. no. MAK135) according to the manufacturer's instructions. Luminescence signals were captured using a TriStar$^2$ S LB 942 microplate reader (Berthold).

**Mitochondrial respiration**. Oxygen consumption rates (OCR) were measured using the XFe24 Extracellular Flux Analyser (AgilentTechnologies, Les Ulis, France). Briefly, culture medium was removed and cells were washed once with DMEM (Sigma-Aldrich), pH buffered at 7.35, supplemented with 10 mM glucose and 2 mM glutamine. Then, cells were incubated with DMEM in CO$_2$-free chamber at 37 °C for 30 min. Baseline OCR was measured: (i) at the basal state, (ii) after oligomycin (Oli) injection (1 μM), (iii) after FCCP injection (0.27 μM and then 0.34 μM), and (iv) after rotenone (Rot)/antimycin A (AA) mix injection (1 μM).

**Assessment of mitochondrial ROS levels**. Mitochondrial ROS levels were measured by incubating cells with MitosoxGreen and Mitotracker Deep Red probes (Life Technologies) for 30 min in culture medium. Cells were washed twice in

phenol-red free culture medium and analysed by confocal microscopy. ROS levels were estimated by calculating the mean fluorescence ratio Mitosox/Mitotracker.

**Animal study**. NOD-SCID mice (NOD.CB17-Prkdscid/NCrHsd) 5 weeks old, male, mean weight) were purchased from Envigo (Gannat, France), maintained in laminar-flow boxes under standard conditions (standard diet and water ad libitum) at 23 °C with 12-h light and 12-h dark cycles at Animal Facility Core (University of Lille, France) under Specific Pathogen Free (SPF) conditions. After one week of quarantine and acclimatisation, the mice were earmarked and randomly separated into 2 groups of 10 subjects, 5 animals/cage, by an independent person. Animal study was approved by the Animal Experimentation Ethics Committee CEEA Nord —Pas de Calais no. 75 (Permit No. 2017062015327270), and all procedures were conducted in accordance with the French National Chart for Animal Experimentation and the ARRIVE guidelines.

To evaluate the effect of TRPC3 down regulation in HPrFs on the proliferative potential of prostate epithelial cancer cells, NOD-SCID mice were injected s.c. on the right flank with a mix of $1 \times 10^6$ PC3-Luc cells and $0.5 \times 10^6$ HPrFs stably expressing doxycycline-inducible shRNA against TRPC3 or scramble sequence (ratio 2:1) and previously treated for 7 days with doxycycline (1 μg ml$^{-1}$) to induce shRNAs expression, in vivo animal imaging was performed one day after injection and weekly by I.P. injection of 150 mg/kg D-XenoLight D-Luciferin–K$^+$ salt bioluminescent Substrate (PerkinElmer, OH, USA) using the IVIS Spectrum optical imaging system (PerkinElmer) for a total of 6 weeks. Bioluminescence recording was performed with the Living Image software version 4.5 from PerkinElmer.

**Public datasets analysis**. Microarray datasets from Oncomine ("Grasso Prostate" and "Tamura Prostate") were used to analyse the expression of TRPC3 in prostate cancer patients undergoing or not chemotherapy or radiotherapy. Log2 median-centred ratios for each patient were used to create the scatter plot.

Raw data (accession number GSE130727 and GSE35988) were downloaded from Gene Expression Omnibus. For GSE130727 dataset, normalisation and differential expression was done with 'DESeq2' R-package version 1.24.0 and R version 3.6.0. Lastly, we extracted normalised counts for TRPC3 on log scale for each condition and treatments (Supplementary Fig. 1l).

For GSE35988 dataset, patients were classified using the median expression of the TRPC3 gene. Differentially expressed genes (DEGs), were identified using the Linear Models for Microarray and RNA-Seq Data limma package v3.46.0 and R 3.6.0. Fold changes were calculated after loess within-normalisation and quantile between-normalisation. Replicate probes were replaced with their average using the limma 'avereps' function. A total of 1585 up-regulated and 1801 down-regulated genes were identified with a cut-off value of absolute log fold change (logFC) > 0.5 and an adjusted $p$-value < 0.05 (Benjamini–Hochberg).

Gene Set Enrichment Analysis (GSEA) was performed using the GSEA software version 4.1.0 from UC San Diego and Broad Institute, using the Human Cellular Senescence Gene Database (HCSGD)[35]. All probe sets were pre-ranked using a metric defined as the sign(logFC)* − log10($p$-value)[62]; thereafter, the ordered probe set list was used as the pre-ranked GSEA input. Detailed GSEA parameters are as follows: the number of permutations is 1000, min size is 15 and max size is 500.

**Statistical analysis**. Error bars represent either the SD or SEM, as described in figure legends. Sample size for each experiment, $n$, is included in the associated figure legend. Statistical significance was determined by the Student's $t$ test and the one-way ANOVA using GraphPad Prism 8.0. $p$ values < 0.05 were considered significant. $p$ values for each experiment are also included in the associated figure legends.

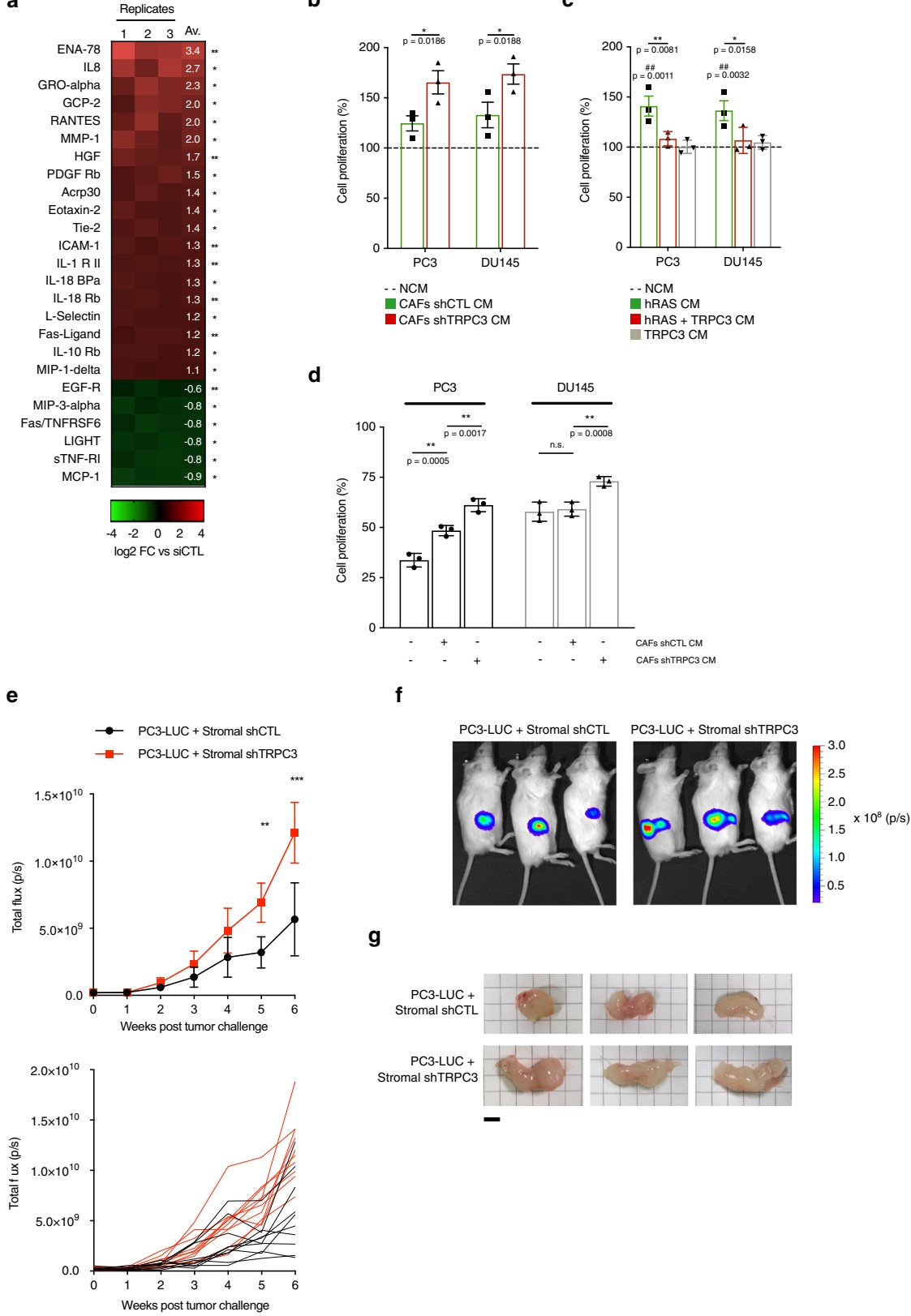

**Fig. 8 TRPC3 knockdown-induced senescence is responsible for the pro-tumour behaviour of prostate stromal cells. a** Heatmap showing the log2 fold change in secreted SASP, as revealed by antibody arrays, by CAFs treated with siTRPC3 and compared to siCTL cells ($n = 3$, multiple $T$-test, two-sided, Holm–Sidak correction for multiple comparisons). **b** Bar diagram plot comparing cell proliferation measured with MTS proliferation assay in PC3 (left) and DU145 (right) cell lines cultured in: (1) non-conditioned medium (NCM), (2) conditioned medium from the siCTL-transfected CAFs (CAFs siCTL CM) and (3) conditioned medium from the siTRPC3-transfected CAFs (CAFs siTRPC3 CM). The readings (light absorption) from PC3 or DU145 cells obtained in CAFs siCTL CM and CAFs siTRPC3 CM were normalised to corresponding readings obtained in NCM and expressed as % (mean ± SEM, $n = 3$: one-way ANOVA, Tukey's multiple comparisons test). **c** Same as in (**b**) but illustrating cell lines cultured in conditioned medium from: (1) control cells, (2) hRAS-overexpressing cells (hRAS), (3) hRAS + TRPC3-overexpressing cells or (4) cells overexpressing TRPC3 alone (mean ± SEM, $n = 3$: one-way ANOVA, Tukey's multiple comparisons test). **d** Bar diagram plot comparing survival of PC3 and DU145 cells (as indicated) after docetaxel treatment between the cell groups described in (**b**). The same measurements were done on the cells exposed to DMSO (vehicle) instead of docetaxel. Chemotherapy-resistance was expressed as a ratio of the cell survival observed after treatment with docetaxel to that observed with vehicle only and expressed in % (mean ± SEM, $n = 3$: one-way ANOVA, Tukey's multiple comparisons test). **e** Time course (6 weeks) of the bioluminescence mean (10 mice; ±SD) total flux (top) and underlying individual traces (bottom) obtained 15 min after luciferin administration. The mice, continuously admitting doxycycline, were injected with PC3-Luc cells mixed with either siCTL-transduced (black line) or siTRPC3#1-transduced (red line) HPrFs. ***$P < 0.001$, **$P < 0.01$ (one-way ANOVA, Tukey's multiple comparisons test). **f** Bioluminescence images of oxidised luciferin from 3 mice of the two groups (panel **e**) on the 6th week after tumour challenge. **g** Comparison of sizes from tumours collected 6 weeks after tumour challenge. Scale bar: 0.5 cm.

**Reporting summary.** Further information on research design is available in the Nature Research Reporting Summary linked to this article.

## Data availability

The processed GSE130727 and GSE35988 datasets used to analyse TRPC3 expression in senescence (Supplementary Fig. 1I–L) are available in Gene Expression Omnibus (GEO) and in Oncomine Database (https://www.oncomine.org/) identified as "Grasso Prostate" and "Tamura Prostate". The dataset used to calculate gene enrichment (Supplementary Fig. 1M) is available on the Human Cellular Senescence Database (HCSGD). The individual data point sets for the remaining figures are available in the Source File provided with this manuscript. Source data are provided with this paper.

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

## Acknowledgements

The authors wish to thank the Cell Imaging and the Flow Core facilities of the BioImaging Center Lille-Nord de France (BICeL) and the Evo-Eco-Paléo Unit—UMR 8198 (University of Lille) for access to the equipment. We thank also Mr. Etienne Dewailly for his technical support. This work was supported by grants from the Institut National de la Santé et de la Recherche Médicale (INSERM), the University of Lille, the Fondation pour la Recherche Médicale (FRM-no. SPF20130526749, V.F.), the Fondation de France (FdF-no. 0057928, V.F.), the GIS ONCOLille and the Institut National du Cancer (INCa) no. PLBIO16-079/2016-153 (D.M., N.P.) and No. 2018-144 (D.B.), the Ligue Nationale Contre le Cancer (N.P.) and the Région Hauts de France (N.P.). The authors would also like to acknowledge the National Council for Scientific Research of Lebanon (CNRS-L), the Agence Universitaire de la Francophonie (AUF), and the Lebanese University (LU) for granting a doctoral fellowship to L.M.

## Author contributions

Conceptualisation, V.F. and N.P.; Methodology, V.F., D.V.G., L.M, E.G., I.F., M.R., D.B., D.G.; B.Q., L.L.; Investigation, V.F., D.V.G., L.M., Y.T., M.L., G.S., O.I., L.L., L.N., W.L., J.K., P.M., N.Z., F.B., M.F.; Writing—original draft, V.F., D.V.G. and N.P.; Writing—review, V.F., D.V.G., L.M, N.P., D.M. D.B., B.Q. and J.B.P.; Funding acquisition, V.F., D.M. and N.P.; Resources, E.D. and L.A.; Supervision, V.F. and N.P.

## Competing interests

The authors disclose a Report of Invention: "Set de oligonucleotides de qPCR pour l'établissement du profil d'expression du domaine fonctionnel des 26 canaux de type TRP" Reference # DSO2018005437), Patent Pending (D.G., M.B., N.P.) and thus the exact primer sequences and positions under declaration of invention cannot be stated. The authors declare no other competing interests.
