## [Peer Review File · Nature Communications]

TRPC3 shapes the ER-mitochondria Ca^{2+} transfer characterizing tumour-promoting senescenceREVIEWER COMMENTS

Reviewer #1 (Remarks to the Author):

This MS proposes that TRPC3 protein acts as a controller of mitochondrial Ca²⁺ load via regulation of IP₃ receptor-mediated Ca²⁺ leak from the endoplasmic reticulum. Expression of TRPC3 is downregulated in senescence causing enhanced cytosolic/mitochondrial Ca²⁺ oscillations and leads to mitochondrial dysfunction characterized by elevated NAD⁺/NADH levels and ROS production. To my knowledge, the studies are novel and potentially important, defining another player and a role for Ca²⁺ signaling in control of cell senescence.

1. Much of the MS concerns the role of TRPC3 in regulation of Ca²⁺ leak from the ER. However, many of these studies appear to be done in proliferating non-senescent cells. The authors should do more to show the relevance of these results in senescent cells and demonstrate their causality of senescence by rescue experiments. For example, do senescent cells show increased Ca²⁺ oscillations? Are these rescued by elevated TRPC3 and does this delay senescence? Is there another way to rescue these oscillations, e.g. with Ca²⁺ chelators, and ask whether this suppresses cell senescence? Similarly, do senescent cells have altered basal mitochondrial load and does manipulation of this affect onset of cell senescence?

2. The elevated basal Ca²⁺ in senescent cells is not apparent in Figure 1J, as stated.

3. Based on mH2A1 ChIP-qPCR, the authors suggest that “accumulation of mH2A1....may be responsible for the reduction of TRPC3 gene expression”. This part of the MS is isolated and not well developed. A causal role for mH2A1 is not shown. Previous studies have mapped mH2A1 across the genome of senescent cells by ChIP-seq. [https://www.cell.com/molecular-cell/pdfExtended/S1097-2765\(15\)00569-9](https://www.cell.com/molecular-cell/pdfExtended/S1097-2765(15)00569-9). Do these studies support specific role of mH2A1 in regulation of TRPC3? The authors might be better to remove this part of the story, if causality is not shown.

4. Figure 2A-C. The authors should show by other molecular markers that TRPC3 knock down induces senescence, e.g. by EdU label, activation of p53-p21 and p16-pRB pathways, DNA damage signaling, downregulation of lamin B1 etc.

5. The authors report “a positive correlation between Trpc3 and the expression of the proliferation marker Ki-67, while a negative correlation was found between Trpc3 and the expression of genes involved in senescence (p27Kip1, SPRY2 and CXCL1)”. This analysis seems very selective. Since they are analyzing a public gene expression dataset, they should look more widely at senescence gene signatures, not individual genes.

6. In Figure S3D, the bands in the IP do not appear to co-migrate with the input. In the IP doublets are apparent and a single band in the input. The identity of these bands should be confirmed by si/shRNA knock down.

7. In Figures 4H, I ectopic expression of WT and mutant TRPC3 should be confirmed by western blot. Throughout the MS ectopic expression and KD should be confirmed.

Reviewer #2 (Remarks to the Author):

The manuscript entitled, “TRPC3 shapes the ER-mitochondria Ca²⁺ transfer characterizing tumour-promoting senescence” is an intriguing and potentially important study. The authors establish that TRPC3 is a negative regulator of IP₃R activity. They further establish that this inhibition contributes to cellular senescence, with clear implications to tumor growth established within the study. Aside from the need for some editing for language and clarity, the primary concern I have is regarding the mechanism of TRP3-mediated IP₃R inhibition. The authors do not provide any data regarding how this occurs although experiments performed in unilamellar

vesicles imply that TRPC3 must be able to inhibit IP3R from within the ER. Is that the only way that this occurs? What is the degree of colocalization between IP3R3 and TRPC3 within the ER under endogenous expression? Answers to these questions would help shed some light on this mechanism and make this relationship less phenomenological.

Specific comments:

1. It's extremely unclear what is being shown in figure 3A. The text should be revised to explain it.
2. There is a general lack of consensus regarding whether or not IP3Rs mediate ER Ca²⁺ leak and it is known that the ER of IP3R-null cells remains leaky. However, perhaps this is a matter of terminology. In figure 4, "ER leak rate" is measured in the presence of different concentrations of IP3. I don't really agree with referring to the Ca²⁺ that passes through a gated channel as "leak".
3. Since TRPC3 inhibits IP3R activity in GUVs, it presumably does so from within the ER. Can the authors show that TRPC3 is present in the ER under endogenous expression?
4. I found figure 8 somewhat confusing. In figure 8B, DOX is used as a chemotherapeutic agent. In figure 8C, doxycycline is used for inducible TRPC3 knockdown. Is DOX doxycycline? Doxorubicin? If it is defined somewhere, I don't see it. Also, concentrations are not listed anywhere.

Minor comments:

1. The text references figure 2E on page 7/8, but there is not figure 2E. This needs to be corrected.

Reviewer #3 (Remarks to the Author):

The study by Farfariello et al. explores the role of TRPC3 loss during tumor-associated senescence. The investigators show that knock-down of TRPC3 expression in fibroblasts mimics the cellular Ca²⁺ changes associated with senescence, linking this to the regulation of the type 3 IP3 Receptor, and altered ER-mitochondrial Ca²⁺ transfer. Conceptually the studies are of interest, as previous work from their group demonstrated that knock-down of all three IP3 Receptors could negatively affect the development of senescence (Wiel et al 2015), but only IP3R2 expression was upregulated transcriptionally during oncogene-induced senescence (OIS). This sets the premise for the current work investigating how the other IP3 Receptors are regulated at the activity level. However, there are several issues that significantly lessen the impact of this work, as described in detail below.

1. Although there is some evidence to demonstrate the role of TRPC3 on IP3R regulation and IP3R mediated Ca²⁺ release, a major concern is that the role of TRPC3 as a regulator of senescence is not strong, as illustrated by several examples below.
2. The authors use several GEO data sets to assess correlations of expression between TRPC3 and senescence markers in tumor tissues. However, it is unclear why the authors did not screen GEO data sets such as GSE26143 or GSE115301, which report gene expression changes in response to treatment with pro-senescence compounds or in response to RAS expression to confirm their findings of decreased TRPC3 expression in senescent cells.
3. The data in Figure 1A need to be presented more rigorously. Replicate data sets should be shown on the heatmap. The selection for TRPC3 as a candidate seems to be a biased selection, rather than based on a statistically-determined cut-off for expression change. For example, one might argue that TRPM1 and TRPM8 are more strongly decreased in response to Dox in the HPrF cells and OIS in MRC-5 cells, and that there is at least a 2 fold increase in some senescence models in TRPC1 & 4. Moreover, the log₂fold change strongly depends on the basal expression levels in untreated cells (this was not shown in the supplemental data), which could explain why some cell culture models show very little change in expression for several of the mRNAs tested.
4. To follow on from point 3 above, since MCU was previously implicated in senescence by the authors it is unclear why this, and other proteins of importance to mitochondrial Ca²⁺

- homeostasis like VDAC, and NCLX were not included in the expression screen in Figure 1A. Moreover, as the authors state in the introduction, the regulation and interaction of IP3Rs is multifaceted (references 22- 25) and it is unclear why a wider screen or an RNAseq screen was not used to test the expression changes of all known interacting partners and regulators of IP3Rs.
5. Since the strongest decrease in TRPC3 expression occurred in the RAS-dependent (OIS) model (Fig 1H) it is unclear why this was not utilized in subsequent experiments and why Ca²⁺ data for this model was not shown as in 1J.
 6. The data in Figure 1J and 2C are purely correlative, and do not prove that TRPC3 loss causes the cellular Ca²⁺ profile of senescence cells. Is it possible to “rescue” with TRPC3 overexpression in this model? In their previous work (Wiel et al 2014), the authors were able to demonstrate that knock-down of the type 2 IP3R can promote OIS escape. One might assume that TRPC3 expression might similarly induce escape from senescence, or at least rescue the mitochondrial Ca²⁺ accumulation induced by OIS or Dox. As such, mitochondrial Ca²⁺ and IP3R-dependent Ca²⁺ leak were not shown in senescent models nor the effects of TRPC3 tested in the context of these senescence model (Figs 3C & 4D). Instead, all data beyond Figure 2B are only carried out in untreated fibroblasts and not in chemotherapy or oncogene-induced senescence models. The authors also do not justify why they decided to only use the Dox inducible senescence model for Figures 1J – 2B, even though the effects of OIS in decreasing TRPC3 expression are probably the strongest.
 7. The data on beta-gal staining in response to TRPC3 knock-down is not strong. % positive beta-gal cells and integrated intensity of staining, as well as statistical analyses of these should be provided. This should be verified by assessing other senescence markers in response to TRPC3 k/d (e.g. suppl Fig 2F).
 8. Based on relative RNA expression, the justification for focusing primarily on type 3 IP3R is not justified, especially in light of the group’s previous findings that all three IP3 Receptors are involved in senescence (Wiel et al 2014) .
 9. The data suggesting that TRPC3 loss causes mitochondrial dysfunction are weak and purely based on ADP/ATP and NAD⁺/NADH ratios, which is not proof of mitochondrial dysfunction. This needs to be demonstrated by measuring respiration or by interrogating the activity of individual ETC complexes. Moreover, an increase in mitoSox staining suggests an increase in superoxide production which can also result as a consequence of increased OxPhos. While mitochondrial dysfunction can induce a senescence phenotype (MiDAS), most studies on OIS demonstrate that senescent cells increase glycolysis, TCA cycle and OxPhos (e.g. Lie et al 2013 Proteomics; Takebayashi et al 2015 Aging Cell; Nacarelli et al., 2019 Nature Cell Biol). In addition, MiDAs was linked to a decrease in the total cellular +NAD/NADH ratio (Wiley et al Cell Metab 2016). These previous studies were not discussed adequately in light of the relatively weak findings related to changes in mitochondrial function in the present work.
 10. How a loss in TRPC3 regulates the SASP is not clear. There is no proof that this is either directly due to TRPC3 loss-mediated change in cellular Ca²⁺ or mitochondrial ROS.
 11. In Figure 8 the effects of conditioned media from TRPC3 k/d fibroblasts on prostate cancer cell proliferation and viability are modest at best. This suggests that the decreases in TRPC3 expression are a likely consequence of senescence but that the knock-down of TRPC3 itself is not sufficient in driving the effects of SASP on tumor cells. Moreover, this figure is missing an important control to demonstrate how the effects of TRPC3 k/d conditioned media compare to conditioned media from OIS or Dox induced senescent fibroblasts.
 12. Fig 8A: Changes in cytokine expression in response to TRPC3 knock-down should be independently verified for the top targets. What does the x axis on the volcano plot represent (Log₂ fold change)?
 13. Throughout the manuscript there are issues with the depiction of data. Some examples are listed: All replicate data points should be superimposed on top of the graphs. Some control blots are missing, including RAS expression in Fig 1H & I. Western blot quantification in Figure 1 should be provide +/- SEM in addition to mean densitometry values. Figure S3 is missing molecular weight markers. Figure legends and panels often do not provide information of the type of fibroblast lines were used in the assays.
 14. Minor: in the text figures 2C & 2D are mislabeled as 2D & 2E.
 15. Minor: line 265: should “detrimental” be replaced with “beneficial”?

Reviewer #1: This MS proposes that TRPC3 protein acts as a controller of mitochondrial Ca²⁺ load via regulation of IP₃ receptor-mediated Ca²⁺ leak from the endoplasmic reticulum. Expression of TRPC3 is downregulated in senescence causing enhanced cytosolic/mitochondrial Ca²⁺ oscillations and leads to mitochondrial dysfunction characterized by elevated NAD⁺/NADH levels and ROS production. To my knowledge, the studies are novel and potentially important, defining another player and a role for Ca²⁺ signaling in control of cell senescence.

> The authors thank the Reviewer for critical comments which have helped us to improve our work. Point-by-point answers to the reviewer comments are given below.

1. Much of the MS concerns the role of TRPC3 in regulation of Ca²⁺ leak from the ER. However, many of these studies appear to be done in proliferating non-senescent cells. The authors should do more to show the relevance of these results in senescent cells and demonstrate their causality of senescence by rescue experiments. For example, do senescent cells show increased Ca²⁺ oscillations?

> Indeed, senescent cells show increased Ca²⁺ oscillations compared to their proliferative counterpart. We already showed increased oscillations in docetaxel (chemotherapy) - induced senescence (former Fig.S1M, new Fig. S2A). We have now included in the MS additional experiments performed on OIS, thus reinforcing our findings (Fig.S2B)

- Are these rescued by elevated TRPC3 and does this delay senescence?

> In our hands, TRPC3 overexpression was not able to decrease the oscillatory pattern observed in senescent cells, meaning that this is more likely a senescence-related rather than a TRPC3-dependent phenotype. To address the second part of this point, we used MRC5-hRAS cells expressing doxycycline-inducible TRPC3. As shown in the new Fig.2F, TRPC3 rescue promotes the escape from senescence.

- Is there another way to rescue these oscillations, e.g. with Ca²⁺ chelators, and ask whether this suppresses cell senescence?

> Indeed, it is possible to manipulate Ca²⁺ oscillations with Ca²⁺ chelators like EGTA or BAPTA-AM. However, given the importance of Ca²⁺ homeostasis in cell physiology, chronic exposure of the cells to such chelators affects cell viability, making impossible to verify their effect on senescence. We are actually investigating and collecting data about the role of Ca²⁺ oscillations in the control of the senescence phenotype but these will be the subject of another paper that will be submitted soon.

- Similarly, do senescent cells have altered basal mitochondrial load and does manipulation of this affect onset of cell senescence?

> To address this point, we used MRC5-hRAS as model of senescence and measured the mitochondrial steady-state Ca²⁺ levels with the mitochondrial-targeted Ca²⁺ probe GECO1-mito. The new results show that senescent cells display a dramatic elevation of basal mitochondrial Ca²⁺ load. TRPC3 rescue (that promotes senescence escape as mentioned above) or the downregulation of the mitochondrial calcium uniporter (MCU), that is already known to promote senescence escape (Wiel et al., Nat. Comm, 2014) prevented the elevation of basal mitochondrial Ca²⁺ load observed in senescent cells (new Fig. 5B and S5D).

The text has been modified accordingly.

2. The elevated basal Ca²⁺ in senescent cells is not apparent in Figure 1J, as stated.

> We agree with the reviewer that the difference on basal Ca²⁺ levels is not really apparent. However, as shown in the new Fig 1G and 1I, there is a significant difference between the basal Ca²⁺ of proliferating and senescent cells. We have thus modified the text with “a slight but significant difference in basal Ca²⁺ levels”.

3. Based on mH2A1 ChIP-qPCR, the authors suggest that “accumulation of mH2A1.... may be responsible for the reduction of TRPC3 gene expression”. This part of the MS is isolated and not well developed. A causal role for mH2A1 is not shown. Previous studies have

mapped mH2A1 across the genome of senescent cells by ChIP-seq. [https://www.cell.com/molecular-cell/pdfExtended/S1097-2765\(15\)00569-9](https://www.cell.com/molecular-cell/pdfExtended/S1097-2765(15)00569-9). Do these studies support specific role of mH2A1 in regulation of TRPC3? The authors might be better to remove this part of the story, if causality is not shown.

> We agree with the reviewer that our findings on the enrichment of the mH2A1 protein on TRPC3 loci are not enough to prove a causal role for TRPC3 downregulation in senescent cells. Following the reviewer's advice, we decided to remove this paragraph from the manuscript.

4. Figure 2A-C. The authors should show by other molecular markers that TRPC3 knock down induces senescence, e.g. by EdU label, activation of p53-p21 and p16-pRB pathways, DNA damage signaling, downregulation of lamin B1 etc.

> We thank the reviewer for this suggestion. We included additional markers to reinforce our findings on TRPC3 knock down-induced senescence. In particular, we added: (1) EdU labelling, (2) WB showing the activation of the p53 and p16 pathways and (3) immunofluorescence showing senescence-associated heterochromatin foci in siTRPC3 cells. The results are shown in the new Fig. 2C-E and the text has been modified accordingly.

5. The authors report "a positive correlation between Trpc3 and the expression of the proliferation marker Ki-67, while a negative correlation was found between Trpc3 and the expression of genes involved in senescence (p27Kip1, SPRY2 and CXCL1)". This analysis seems very selective. Since they are analyzing a public gene expression dataset, they should look more widely at senescence gene signatures, not individual genes.

> We agree with the reviewer that the selected genes may not represent a senescent signature. To address this point, we performed a GSEA analysis on the GSE35988 dataset, testing the enrichment of genes upregulated in senescence in tumour samples showing different degrees of TRPC3 expression. As shown in the new Fig.S1M, the genes upregulated in senescence show a high and significant degree of enrichment among the genes negatively correlating with TRPC3 expression, confirming the hypothesis that TRPC3 expression is negatively correlated with senescence signatures. The text has been modified according to these new results.

6. In Figure S3D, the bands in the IP do not appear to co-migrate with the input. In the IP doublets are apparent and a single band in the input. The identity of these bands should be confirmed by si/shRNA knock down.

> We thank the reviewer for this suggestion. We think that the difference in migration originates from a problem in the input lane. We have substituted the figure with another replicate of the Co-IP. Please find the uncropped gel below.

7. in Figures 4H, I ectopic expression of WT and mutant TRPC3 should be confirmed by western blot. Throughout the MS ectopic expression and KD should be confirmed.

> We thank the reviewer for this suggestion. The ectopic expression of TRPC3 is now shown in Fig. S4D-E (mutants) and Fig. S2I (overexpression of WT TRPC3 and hRAS).

Reviewer #2: The manuscript entitled, “TRPC3 shapes the ER-mitochondria Ca²⁺ transfer characterizing tumour-promoting senescence” is an intriguing and potentially important study. The authors establish that TRPC3 is a negative regulator of IP₃R activity. They further establish that this inhibition contributes to cellular senescence, with clear implications to tumour growth established within the study. Aside from the need for some editing for language and clarity, the primary concern I have is regarding the mechanism of TRPC3-mediated IP₃R inhibition. The authors do not provide any data regarding how this occurs although experiments performed in unilamellar vesicles imply that TRPC3 must be able to inhibit IP₃R from within the ER. Is that the only way that this occurs? What is the degree of colocalization between IP₃R and TRPC3 within the ER under endogenous expression? Answers to these questions would help shed some light on this mechanism and make this relationship less phenomenological.

> We thank the reviewer for his critical reading of our manuscript. Point-by-point answers to the comments are provided below.

Specific comments:

1. It's extremely unclear what is being shown in figure 3A. The text should be revised to explain it.

> We have modified the Figure 3A and the corresponding text to make it clearer to the readers.

2. There is a general lack of consensus regarding whether or not IP₃R_s mediate ER Ca²⁺ leak and it is known that the ER of IP₃R-null cells remains leaky. However, perhaps this is a matter of terminology. In figure 4, “ER leak rate” is measured in the presence of different concentrations of IP₃. I don't really agree with referring to the Ca²⁺ that passes through a gated channel as “leak”.

> Even though there is growing evidence for spontaneous activity of PLC driving background activity of IP₃R_s that contributes to Ca²⁺ leak from the S/ER in many cell types (e.g. Gordienko et al (2015). *Cardiovascular Research* **105**:131-142), we agree that historically the process of Ca²⁺ efflux from the S/ER in vast majority of cases is referred to as IP₃R-mediated Ca²⁺ release. Taking into account your concerns, we have modified the text of the manuscript accordingly.

3. Since TRPC3 inhibits IP₃R activity in GUVs, it presumably does so from within the ER. Can the authors show that TRPC3 is present in the ER under endogenous expression?

> To address this point, we have performed a subcellular fractionation followed by WB to assess TRPC3 expression in the different fractions, and included the results in the new Supplementary Fig.S3D. The results show that TRPC3 is expressed in the plasma membrane and the ER fraction, meaning that TRPC3 can exert its inhibitory functions towards the IP₃R_s within the ER.

4. I found figure 8 somewhat confusing. In figure 8B, DOX is used as a chemotherapeutic agent. In figure 8C, doxycycline is used for inducible TRPC3 knockdown. Is DOX doxycycline? Doxorubicin? If it is defined somewhere, I don't see it. Also, concentrations are not listed anywhere.

> We agree with the reviewer that the nomenclature might be somewhat confusing. We have modified the legends in the text (Docetaxel = chemotherapeutic agent to induce senescence, DOX = doxycycline to induce conditional expression of shRNAs or TRPC3). We added the description of the different compounds as well as the concentration used in the corresponding figure legends.

Minor comments:

1. The text references figure 2E on page 7/8, but there is not figure 2E. This needs to be corrected.

> We thank the reviewer for this comment and we have corrected the mistake.

Reviewer #3: The study by Farfariello et al. explores the role of TRPC3 loss during tumor-associated senescence. The investigators show that knock-down of TRPC3 expression in fibroblasts mimics the cellular Ca²⁺ changes associated with senescence, linking this to the regulation of the type 3 IP3 Receptor, and altered ER-mitochondrial Ca²⁺ transfer. Conceptually the studies are of interest, as previous work from their group demonstrated that knock-down of all three IP3 Receptors could negatively affect the development of senescence (Wiel et al 2015), but only IP3R2 expression was upregulated transcriptionally during oncogene-induced senescence (OIS). This sets the premise for the current work investigating how the other IP3 Receptors are regulated at the activity level. However, there are several issues that significantly lessen the impact of this work, as described in detail below.

> We thank the reviewer for obvious care and critical comments for our manuscript. Point-by-point answers to the comments are given below.

1. Although there is some evidence to demonstrate the role of TRPC3 on IP3R regulation and IP3R mediated Ca²⁺ release, a major concern is that the role of TRPC3 as a regulator of senescence is not strong, as illustrated by several examples below.

2. The authors use several GEO data sets to assess correlations of expression between TRPC3 and senescence markers in tumor tissues. However, it is unclear why the authors did not **screen GEO data sets such as GSE26143 or GSE115301, which report gene expression changes in response treatment with pro-senescence compounds or in response to RAS expression to confirm their findings of decreased TRPC3 expression in senescent cells.**

> We agree with the reviewer that the analysis of TRPC3 expression in datasets would strengthen the correlation between TRPC3 expression and senescence. To address this point, we analysed TRPC3 expression in the dataset GSE130727, reporting gene expression changes in response to hRAS activation or doxorubicin treatment in WI-38 fibroblasts. In addition, to reinforce our findings on TRPC3 expression and senescence, we performed a GSEA analysis on the GSE35988 dataset, testing the enrichment of genes upregulated in senescence in tumour samples showing different degrees of TRPC3 expression. As shown in the revised version of our manuscript, the genes upregulated in senescence show a high and significant degree of enrichment among the genes negatively correlating with TRPC3 expression, confirming the hypothesis that the level of TRPC3 expression is negatively correlated with senescence signatures. The results have been added in the new Fig. S1L-M and the text has been modified accordingly.

3. **The data in Figure 1A need to be presented more rigorously. Replicate data sets should be shown on the heatmap.** The selection for TRPC3 as a candidate seems to be a biased selection, rather than based on a statistically-determined cut-off for expression change. For example, one might argue that TRPM1 and TRPM8 are more strongly decreased in response to Dox in the HPrF cells and OIS in MRC-5 cells, and that there is at least a 2 fold increase in some senescence models in TRPC1 & 4. Moreover, the log₂fold change strongly depends on the basal expression levels in untreated cells (this was not shown in the supplemental data), which could explain why some cell culture models show very little change in expression for several of the mRNAs tested.

> We thank the reviewer for this comment. The selection for TRPC3 as a candidate was determined by the fact that its downregulation was consistent and statistically significant within all the senescence models, and showed the highest mean degree of difference in terms of fold changes compared to the controls, among the other genes analysed. We agree with the reviewer that the representation of the data on the heatmap may evoke certain concerns about the method used to choose TRPC3 as candidate. To address this point, we have modified the heatmap in Fig. 1A that is now showing triplicates and the mean values of the fold change for each senescence model, as well as statistical significance. In addition, we are now showing the mean Ct values of the genes in untreated cells in Suppl. Table 1 and

the statistical analysis in Suppl. Table 2. We modified the text and the figure legends accordingly, clarifying the rationality of our choice of TRPC3 as a candidate. As we have explained in the text, we determined a cut-off for expression change at +/- 1.5 folds (vs CTL) and we looked at the genes whose expression was modulated consistently among the senescence models. To reduce the redundancy of the results (we are now showing the replicates for each gene tested), we removed the former panels B, D, F and H from Fig.1.

4. To follow on from point 3 above, since MCU was previously implicated in senescence by the authors it is **unclear why** this, and **other proteins of importance to mitochondrial Ca²⁺ homeostasis like VDAC, and NCLX were not included in the expression screen in Figure 1A**. Moreover, as the authors state in the introduction, the regulation and interaction of IP3Rs is multifaceted (references 22- 25) and it is unclear why a wider screen or an RNAseq screen was not used to test the expression changes of all known interacting partners and regulators of IP3Rs.

> We thank the reviewer for this comment. Following his advice and to give a comprehensive view of the Ca²⁺ channels implicated in the ER-mitochondrial homeostasis, we added VDAC, NCLX and MCU expression to the heatmap in Fig.1A.

Concerning the possibility to do a RNAseq screening to check for all known IP3R interacting proteins, we agree with the reviewer that it might have been an alternative to qPCR screening. However, in our opinion this method would require a lot of efforts and considerable time for the analysis, generating a massive amount of data that would deserve to be treated as a separate subject.

5. Since the strongest decrease in TRPC3 expression occurred in the RAS-dependent (OIS) model (Fig 1H) it is unclear why this was not utilized in subsequent experiments and why Ca²⁺ data for this model was not shown as in 1J.

> We thank the reviewer for this comment. To increase the relevance of our findings, we added Ca²⁺ data for the OIS model in the new Fig. 1 (panels H-I).

6. The data in Figure 1J and 2C are purely correlative, and do not prove that TRPC3 loss causes the cellular Ca²⁺ profile of senescence cells. **Is it possible to “rescue” with TRPC3 overexpression in this model?** In their previous work (Wiel et al 2014), the authors were able to demonstrate that knock-down of the type 2 IP3R can promote OIS escape. One might assume that TRPC3 expression might similarly induce escape from senescence, or at least **rescue the mitochondrial Ca²⁺ accumulation induced by OIS or Dox**. As such, mitochondrial Ca²⁺ and IP3R-dependent Ca²⁺ leak were not shown in senescent models nor the effects of TRPC3 tested in the context of these senescence model (Figs 3C & 4D). Instead, all data beyond Figure 2B are only carried out in untreated fibroblasts and not in chemotherapy or oncogene-induced senescence models. The authors also do not justify why they decided to only use the Dox inducible senescence model for Figures 1J – 2B, even though the effects of OIS in decreasing TRPC3 expression are probably the strongest.

> To address this point, we transformed MRC5-hRAS with lentiviral particles bearing doxycycline-inducible TRPC3. As shown in the new Fig.2F and 5B, TRPC3 rescue is able to (1) prevent the mitochondrial Ca²⁺ accumulation occurring in senescent cells and (2) to promote senescence escape, as shown by EdU labelling and cell density/morphology. In addition, we are now showing the Ca²⁺ signature for OIS in Fig.1H and Fig.S2B.

7. The data on beta-gal staining in response to TRPC3 knock-down is not strong. % positive beta-gal cells and **integrated intensity of staining, as well as statistical analyses of these should be provided**. This should be verified by **assessing other senescence markers in response to TRPC3 k/d** (e.g. suppl Fig 2F).

> We think that this comment is due to the poor quality of the image shown in the old version of the manuscript. We ameliorated the images and, as suggested by the reviewer, we are now showing the % of SA-βgal positive cells in the new Fig.2B and Fig. S2E. In addition, we included supplementary markers to reinforce our findings on TRPC3 KD-induced

senescence. In particular, we have added: (1) EdU labeling, (2) WB showing the activation of the p53 and p16 pathways and (3) immunofluorescence showing senescence-associated heterochromatin foci in siTRPC3 cells (new Fig. 2C-E). The text has been modified by including the new data.

8. Based on relative RNA expression, **the justification for focusing primarily on type 3 IP3R is not justified, especially in light of the group's previous findings that all three IP3 Receptors are involved in senescence** (Wiel et al 2014).

> The choice to focus primarily on type 3 IP3R is due to its predominant expression in comparison to other types, as detected by absolute and not relative qRT-PCR, showed in Fig. S3C and depicting IP₃Rs RNA levels at N of copies/ng of RNA.

9. The **data suggesting that TRPC3 loss causes mitochondrial dysfunction are weak and purely based on ADP/ATP and NAD⁺/NADH ratios, which is not proof of mitochondrial dysfunction**. This needs to be demonstrated by **measuring respiration or by interrogating the activity of individual ETC complexes**. Moreover, an increase in mitoSox staining suggests an increase in superoxide production which can also result as a consequence of increased OxPhos. While mitochondrial dysfunction can induce a senescence phenotype (MiDAS), most studies on OIS demonstrate that senescent cells increase glycolysis, TCA cycle and OxPhos (e.g. Lie et al 2013 Proteomics; Takebayashi et al 2015 Aging Cell; Nacarelli et al., 2019 Nature Cell Biol). In addition, MiDAs was linked to a decrease in the total cellular +NAD/NADH ratio (Wiley et al Cell Metab 2016). These previous studies were not discussed adequately in light of the relatively weak findings related to changes in mitochondrial function in the present work.

> We agree with the reviewer that the data provided in the old version of the manuscript could not be a proof of mitochondrial dysfunction following TRPC3 knockdown.

Since ADP/ATP and NAD⁺/NADH ratios were measured on total cell lysates, we decided to perform additional experiments to show ATP mitochondrial levels in siCTL and siTRPC3-treated fibroblasts by using the mitoGO-ATeam fluorescent sensor. The results, shown in the new Fig. 7A, indicate that TRPC3 knockdown induces an increase of the mitochondrial ATP levels.

Moreover, we compared OCR in cells transfected with siCTL, siTRPC3 or the combination of siTRPC3/siIP3R3. In the new Fig.7B-F we are now showing that TRPC3 knockdown triggers an alteration of the mitochondrial metabolism characterized by increasing OCR and OXPHOS, a result completely reverted by the inhibition of IP₃R3 (an additional proof of the predominant role of IP₃R3, as commented in the point 8). This indicates that the increased ER-mito Ca²⁺ transfer caused by TRPC3 loss augments mitochondrial respiration, leading to higher ROS production and senescence.

The figures, figures legends and text were modified in accordance with these new results and the discussion was implemented adequately.

10. How a loss in TRPC3 regulates the SASP is not clear. There is no proof that this is either directly due to TRPC3 loss-mediated change in cellular Ca²⁺ or mitochondrial ROS.

> We thank the reviewer for this comment. Our hypothesis is that TRPC3 loss promotes senescence and that senescent cells produce SASP as they are expected to. We do not think that TRPC3 is directly regulating the SASP. However, we are actually investigating and collecting data on the role of the specific Ca²⁺ signature of senescent cells in the SASP regulation. This will be the subject of another paper that will be submitted soon. To avoid any misinterpretation of the results, we have modified the paragraph title "Altered mitochondrial function occurring upon TRPC3 downregulation is characterized by a tumour-promoting SASP" with "TRPC3 downregulation is characterized by a tumour-promoting SASP".

11. In Figure 8 the effects of conditioned media from TRPC3 k/d fibroblasts on prostate cancer cell proliferation and viability are modest at best. This suggests that the decreases in TRPC3 expression are a likely consequence of senescence but that the knock-down of

TRPC3 itself is not sufficient in driving the effects of SASP on tumour cells. Moreover, this figure is missing an important control to demonstrate how the effects of TRPC3 k/d conditioned media compare to conditioned media from OIS or Dox induced senescent fibroblasts.

> We think that this comment is due to our error in the manuscript text. In page 11, lines 273-275, we stated that “shTRPC3 CAFs increased the proliferation rate of the epithelial cells by about 15% relative to that observed with shCTL CAFs (Fig. 8B)”. However, when looking at the graph and comparing the cell proliferation rates, shTRPC3 cells-conditioned medium increased cell proliferation by about 35% respect to the shCTL cells. This is in line with other’s findings (Bavik et al. Cancer Res 2006 (66) (2) 794-802), showing the effect of senescent cell conditioned medium on the proliferation of prostate cancer cells. However, for comparison, we added in the figure 2C the data relative to cell proliferation induction by conditioned medium from OIS-induced senescent fibroblasts. Of note, TRPC3 rescue almost totally prevented the effects of conditioned medium from senescent cells. The same mistake was introduced for the resistance to chemotherapy (Fig.8C), where the increase of resistance to chemotherapy is about 25 %. In addition, our in vitro results are supported by the in vivo experiments that clearly show the effects of TRPC3 downregulation in stromal cells towards cancer epithelial cells proliferation.

We have thus modified the text accordingly.

12. Fig 8A: Changes in cytokine expression in response to TRPC3 knock-down should be independently verified for the top targets. What does the x axis on the volcano plot represent (Log2 fold change)?

> We thank the reviewer for this comment. We decided to make the Fig.8A clearer to the readers and replaced the volcano plot with a heatmap showing the secretome components that were significantly modulated upon TRPC3 downregulation. Following the reviewer’s advice for the heatmap in Fig. 1A, we are now showing replicates and statistics for each protein (see Fig. 8A of the revised version of the manuscript).

13. Throughout the manuscript there are issues with the depiction of data. Some examples are listed: All replicate data points should be superimposed on top of the graphs. Some control blots are missing, including RAS expression in Fig 1H & I. Western blot quantification in Figure 1 should be provide +/- SEM in addition to mean densitometry values. Figure S3 is missing molecular weight markers. Figure legends and panels often do not provide information of the type of fibroblast lines were used in the assays.

> We thank the reviewer for this comment. We modified the text and figures according to the reviewer’s concern.

14. Minor: in the text figures 2C & 2D are mislabeled as 2D & 2E.

> We have modified the text according to this remark.

15. Minor: line 265: should “detrimental” be replaced with “beneficial”?

> We have modified the text according to this remark.

REVIEWER COMMENTS

Reviewer #1 (Remarks to the Author):

The authors have addressed my concerns and the MS is now acceptable for publication.

Reviewer #3 (Remarks to the Author):

The authors were somewhat receptive to the reviewer's comments by clarifying some of the results and altering the presentation of data. However, several comments from all three reviewers were in my opinion not adequately addressed, or addressed with data that do not support the authors' conclusions, which are at times over-interpreted in the manuscript. I only highlight a few examples here.

An example of a major reviewer comment that was not adequately addressed relates to rescue by TRPC3 expression on Ca²⁺ and senescence phenotypes. The data in 2F are not convincing that TRPC3 overexpression leads to senescence escape, as indicated in the abstract. There is no data to show that overexpression of TRPC3 rescues the SASP.

Effects of histamine in Fig 2H are modest and Fig 2SB does not seem to support the notion that Ca²⁺ release is increased in the additional senescence models shown (OIS). This needs to be addressed in the text and highlighted that cellular Ca²⁺ changes differ between senescence models. Moreover, the different consequences on SOCE are not addressed. Does this also indicate that their conclusions related to the contribution of SOCE to oscillations does not hold true for the OIS model?

The statement in the abstract: "Here, we identify the molecular mechanism underlying the alteration of mitochondrial function in stress-induced senescent fibroblasts" is not justified by the data. There is no direct evidence that senescence is caused by the mitochondrial changes resulting from TRPC3 loss. For example, does mitochondrial superoxide scavenging reverse the senescence in TRPC3 knock-down cells.

The data are not very convincing that there is a strong effect on mitochondrial function with TRPC3 loss. It is unclear how statistical significance was reached looking at individual data points between groups. The term "OCR ATP turnover" is not accurate to describe OCR after oligomycin inhibition. Seahorse experiments are not performed in the senescence model to demonstrate a similar phenotype to TRPC3 knock-down.

Figure S3D still seems to have issues with band migrating at a different MW than the pull down, and justification that this is due to migration in the gel is not adequate.

Some comments were dismissed, such as the need for validation of SASP target genes following TRPC3 knock-down (Fig 8A). Fig 8B-D do not show individual data points superimposed on graphs.

Reviewer #3 (Remarks to the Author):

” The authors were somewhat receptive to the reviewer’s comments by clarifying some of the results and altering the presentation of data. However, several comments from all three reviewers were in my opinion not adequately addressed, or addressed with data that do not support the authors’ conclusions, which are at times over-interpreted in the manuscript. I only highlight a few examples here.

An example of a major reviewer comment that was not adequately addressed relates to rescue by TRPC3 expression on Ca²⁺ and senescence phenotypes ”

We would like to thank the Reviewer for taking the time to provide additional helpful comments allowing further improvement of the manuscript. We are also particularly pleased that our first revised version was well received by expert reviewers 1 and 2 who were completely satisfied after the first round of revision.

We addressed all the criticisms of the Reviewer and hope to modify the interpretation of our results sufficiently to avoid any over-interpretation.

“ The data in 2F are not convincing that TRPC3 overexpression leads to senescence escape, as indicated in the abstract “

> To provide more convincing readouts of senescence escape promoted by TRPC3 rescue we conducted additional experiments to assess p16, PCNA and Cyclin D1 expression and performed SA-β-gal staining and crystal violet assessment of colony formation. We have removed EdU labeling (former Fig. 2F) and added new results Fig.2 (new panels F and G). The figure legend has been modified accordingly.

“ There is no data to show that overexpression of TRPC3 rescues the SASP. “

> We agree with the Reviewer that these results significantly strengthen the physiological role of TRPC3 in the regulation of the senescent phenotype. To address this point, we have performed a qRT-PCR screening on the main SASP elements secreted during OIS (please see for the reference Nacarelli et al. 2019, Fig. 2) and compared gene expression between MRC5 wt, MRC5 -hRAS and MRC5-hRAS overexpressing TRPC3. As it is shown now in the revised Fig. S7B, TRPC3 rescue was able to reduce significantly the level of most of SASP elements, thus, confirming involvement of TRPC3 in regulation of the senescent secretome. The text, figures and figure legends have been modified accordingly.

“ Effects of histamine in Fig 2H are modest and Fig 2SB does not seem to support the notion that Ca²⁺ release is increased in the additional senescence models shown (OIS). This needs to be addressed in the text and highlighted that cellular Ca²⁺ changes differ between senescence models. “

> We apologize if these data were not presented clearly enough. We respectfully point out that the Reviewer may have missed that, according to Fig1., effects of histamine-mediated Ca²⁺ release on OIS are almost doubled relative to the non-senescent counterpart (please see Fig. 11 central panel). However, as noticed by the Reviewer, the amplitude and kinetics of Ca²⁺ signaling could vary according to different senescence modes and, as required by the reviewer, we addressed in the text this important issue.

“ Moreover, the different consequences on SOCE are not addressed. Does this also indicate that their conclusions related to the contribution of SOCE to oscillations does not hold true for the OIS model? ”

> We thank the Reviewer for raising this important question and we certainly understand the value of presenting the different consequences on SOCE. However, the molecular nature of SOCE in fibroblasts and its involvement in oscillations (and in particular in MRC5 cells) is poorly understood and not explored yet. Potentially, SOCE amplitude, kinetics of activation and inactivation of SOC current, may contribute to Ca²⁺ oscillations, but to date no detailed investigation of the mechanisms was reported on SOCE and oscillations in those cell models. Indeed, we have shown in OIS that SOCE is reduced but not abolished, so that Ca²⁺ entry that still occurs in OIS could be sufficient to sustain Ca²⁺ oscillations observed in these cells. The

detailed analysis of the mechanisms of Ca²⁺ oscillations in senescence and the molecular nature of SOCE is a subject for a separate extensive study and is under current investigation in our laboratory. Anyway, to avoid unnecessary speculations and misinterpretations of the data, we have removed the sentence "...which are facilitated by SOCE induction" (page 7).

" The statement in the abstract: "Here, we identify the molecular mechanism underlying the alteration of mitochondrial function in stress-induced senescent fibroblasts" is not justified by the data. There is no direct evidence that senescence is caused by the mitochondrial changes resulting from TRPC3 loss. For example, does mitochondrial superoxide scavenging reverse the senescence in TRPC3 knock-down cells."

> We thank the Reviewer for raising this important issue, although we confess being somewhat perplexed by this comment. Indeed, the effects of mitochondrial superoxide scavenging on senescence induced by TRPC3 knock-down have been shown starting from the first submission of our manuscript. These results are still presented in Fig. S6H, demonstrating strong reduction of the number of SA-β-Gal positive cells following treatment with MitoTempo.

" The data are not very convincing that there is a strong effect on mitochondrial function with TRPC3 loss. It is unclear how statistical significance was reached looking at individual data points between groups. The term "OCR ATP turnover" is not accurate to describe OCR after oligomycin inhibition. Seahorse experiments are not performed in the senescence model to demonstrate a similar phenotype to TRPC3 knock-down."

> In the source data file provided, we have reported single readings from seahorse experiments in the source data file. We believe that this data file (used for statistical analysis) is accessible to the reviewer. In addition, to provide a clearer demonstration of the role of TRPC3 in OCR, we realized additional experiments demonstrating that TRPC3 rescue dramatically decreases OCR in OIS, what confirms further our previous statement. These results have been added to Fig. S7 and the text has been modified accordingly.

With regards to point about the term "ATP turnover", we do not agree with the reviewer comment. During the first three measurements, the basal OCR was determined (from t0min to t15min). This represents the sum of all the oxidative processes consuming O₂ such as the mitochondrial activity of cytochrome c oxidases (Cox IV) as well as the activity of other oxidases (e.g.: NADPH oxidase). However, the decrease in OCR following treatment with oligomycin A represents solely the effect of inhibition of the mitochondrial respiration generating ATP (from t20min to t26min). The remaining respiration known as "proton leak" is linked to other oxygen-consuming mitochondrial processes independent of ATP production.

"- Figure S3D still seems to have issues with band migrating at a different MW than the pull down, and justification that this is due to migration in the gel is not adequate."

> We have repeated Co-IP and updated Fig. S3E accordingly.

"- Some comments were dismissed, such as the need for validation of SASP target genes following TRPC3 knock-down (Fig 8A). "

We have conducted qRT-PCR assays to validate the modulation of SASP target genes upon TRPC3 knock down. As shown in the revised Fig. S8, most of the SASP genes were confirmed by qRT-PCR. The text and figure legends have been modified accordingly.

"- Fig 8B-D do not show individual data points superimposed on graphs."

> In the revised version, Fig. 8B-D show individual data points superimposed on graphs.

REVIEWERS' COMMENTS

Reviewer #3 (Remarks to the Author):

I appreciate that the authors have further clarified my concerns and provided additional data related to SASP target gene validation following knock-down and changes in mRNA levels in response to TRPC3 over expression (Fig S8), as well as providing additional confirmation of senescence markers.

My remaining concerns are the discrepancies associated with mitochondrial respiration related to the data following TRRPC3 over expression in the hRAS model. Quantification of ATP turnover, OCR in response to oliomycin are not provided in Figure S7. Based on the graph in panel A, it appears that hRAS+ TRPC3 leads to an increase in OCR - ATP turn over, compared to hRAS alone. Moreover, the huge increase in respiratory reserve is not mentioned nor explained.

The other concern is the modest effect on proliferation in Figure 8B. Now that individual data points have been superimposed, I do not see how significance was reached between the different experimental groups (raw data values were not provided for Fig 8 in the accompanying excel spread sheet to assess this).

While there is a significant change in in vivo tumor growth, the effects on proliferation in cell culture studies are modest. I don't think mentioning this negatively influences the overarching message of the paper. I just wish the authors would acknowledge this instead of over interpreting the proliferation data of the in vitro studies. There may be other factors that contribute to differences observed in in vivo tumor growth that are not directly related to cancer cell proliferation.

Reviewer #3 (Remarks to the Author):

“I appreciate that the authors have further clarified my concerns and provided additional data related to SASP target gene validation following knock-down and changes in mRNA levels in response to TRPC3 over expression (Fig S8), as well as providing additional confirmation of senescence markers”.

We thank the Reviewer for the additional helpful comments. We hope that modifications, introduced to the revised manuscript in accordance with the Reviewer’s criticisms, eliminate any possible misinterpretation of our results.

“My remaining concerns are the discrepancies associated with mitochondrial respiration related to the data following TRRPC3 over expression in the hRAS model. Quantification of ATP turnover, OCR in response to oligomycin are not provided in Figure S7. Based on the graph in panel A, it appears that hRAS+ TRPC3 leads to an increase in OCR - ATP turn over, compared to hRAS alone. Moreover, the huge increase in respiratory reserve is not mentioned nor explained”.

We thank the Reviewer for this comment. Indeed, compared to normal fibroblast, OIS increased Basal OCR, Proton leak (respiration under oligomycin treatment), maximal OCR, but not ATP turnover respiration (oligomycin-sensitive respiration) (please see new Fig S7). This profile was expected and has been already observed in senescent hepatic Alpha Mouse Liver 12 (AML12) cells treated by H₂O₂ (Singh BK, Tripathi M, Sandireddy R, Tikno K, Zhou J, Yen PM. Decreased autophagy and fuel switching occur in a senescent hepatic cell model system. Aging (Albany NY). 2020; 12:13958-13978. <https://doi.org/10.18632/aging.103740>)

In our study, TRPC3 rescue in OIS decreased both basal OCR and proton leak, but maintained high level of maximal OCR. Maximal respiration is regulated by many processes including level of activity in complexes I, II or III, alteration of complex IV enzymatic activity by oxidation and assembly of respiratory chain components in larger structures, such as the respiratory supercomplexes (please see our review Marchetti P et al., Mitochondrial spare respiratory capacity: Mechanisms, regulation, and significance in non-transformed and cancer cells. FASEB J. 2020 Oct;34(10):13106-13124. doi: 10.1096/fj.202000767R). Upregulation of Ras in association with downregulation of TRPC3 could lead to unexpected regulation of maximal OCR.

Following the reviewer comment, we added quantification of ATP turnover in the new Fig. S7 and modified the manuscript accordingly.

“The other concern is the modest effect on proliferation in Figure 8B. Now that individual data points have been superimposed, I do not see how significance was reached between the different experimental groups (raw data values were not provided for Fig 8 in the accompanying excel spread sheet to assess this).

While there is a significant change in in vivo tumor growth, the effects on proliferation in cell culture studies are modest. I don't think mentioning this negatively influences the overarching message of the paper. I just wish the authors would acknowledge this instead of over interpreting the proliferation data of the in vitro studies. There may be other factors that contribute to differences observed in in vivo tumor growth that are not directly related to cancer cell proliferation”.

We thank the Reviewer for this well spotted point. We apologize for an inaccuracy in the presentation of statistical analysis of the results presented in Fig. 8B. We are now showing p-values and have added corresponding data in the source data file, as it was suggested by the Reviewer. Even though we have found that the difference between the corresponding groups is still significant ($p = 0.0186$ for PC3 cells and $p = 0.0188$ for DU145), we agree with the Reviewer that it is quite modest in comparison with that observed *in vivo*, and, hence, significant change in tumor growth *in vivo* cannot be explained solely by the effects of TRPC3-related SASP on cancer cell proliferation in culture and may recruit some other mechanisms also. The Discussion was changed accordingly.